# A supramolecular photosensitizer derived from an Arene-Ru(II) complex self-assembly for NIR activated photodynamic and photothermal therapy

Gang Xu[1], Chengwei Li[1], Chen Chi[2], Luyan Wu[2], Yanyan Sun[3], Jian Zhao [1✉], Xing-Hua Xia [2✉] & Shaohua Gou [1✉]

Effective photosensitizers are of particular importance for the widespread clinical utilization of phototherapy. However, conventional photosensitizers are usually plagued by short-wavelength absorption, inadequate photostability, low reactive oxygen species (ROS) quantum yields, and aggregation-caused ROS quenching. Here, we report a near-infrared (NIR)-supramolecular photosensitizer (RuDA) via self-assembly of an organometallic Ru(II)-arene complex in aqueous solution. RuDA can generate singlet oxygen ($^1O_2$) only in aggregate state, showing distinct aggregation-induced $^1O_2$ generation behavior due to the greatly increased singlet-triplet intersystem crossing process. Upon 808 nm laser irradiation, RuDA with excellent photostability displays efficient $^1O_2$ and heat generation in a $^1O_2$ quantum yield of 16.4% (FDA-approved indocyanine green: $\Phi_\Delta = 0.2\%$) together with high photothermal conversion efficiency of 24.2% (commercial gold nanorods: 21.0%, gold nanoshells: 13.0%). In addition, RuDA-NPs with good biocompatibility can be preferably accumulated at tumor sites, inducing significant tumor regression with a 95.2% tumor volume reduction in vivo during photodynamic therapy. This aggregation enhanced photo-dynamic therapy provides a strategy for the design of photosensitizers with promising photophysical and photochemical characteristics.

[1] Jiangsu Province Hi-Tech Key Laboratory for Biomedical Research and Pharmaceutical Research Center, School of Chemistry and Chemical Engineering, Southeast University, Nanjing, China. [2] State Key Lab of Analytical Chemistry for Life Science, School of Chemistry and Chemical Engineering, Nanjing University, Nanjing, China. [3] School of Chemistry and Life Sciences, Suzhou University of Science and Technology, Suzhou, China. ✉email: zhaojianzhaokuan@163.com; xhxia@nju.edu.cn; sgou@seu.edu.cn

Compared with conventional therapeutics, photodynamic therapy (PDT) is an appealing cancer therapeutic modality owing to its distinct advantages such as precise spatio-temporal control, non-invasiveness, negligible drug resistance, and minimized side effects[1–3]. Upon light irradiation, photosensitizers used could be activated to generate high reactive oxygen species (ROS), leading to cell apoptosis/necrosis or immune responses[4,5]. However, most conventional photosensitizers, such as chlorins, porphyrins, and anthraquinones, have relatively short-wavelength absorption (frequency < 680 nm), thus resulting in poor light penetration because of the intense absorption of biological molecules (e.g., hemoglobin and melanin) in the visible region[6,7]. As a result, near-infrared (NIR)-absorbing photosensitizers, activated in the "therapeutic window" of 700–900 nm, are highly desired for phototherapy. Since NIR light is absorbed minimally by the biological tissues, it can lead to deeper penetration and less photodamage[8,9].

Unfortunately, the existing NIR-absorbing photosensitizers are usually subject to poor photostability, low singlet oxygen ($^1O_2$) generation capabilities and aggregation-caused $^1O_2$ quenching, which have limited their clinical application[10,11]. Although great efforts have been made to improve the photophysical and photochemical properties of the conventional photosensitizers, NIR-absorbing photosensitizers have seldom been reported to solve all these issues so far. In addition, photosensitizers have hardly been found to hold promise for effectively generating $^1O_2$ under the irradiation of light wavelength above 800 nm[12–14], because the photon energy decreases quickly in the NIR region. Donor-acceptor (D-A) type dyes with triphenylamine (TPA) as electron-donor species and [1,2,5]thiadiazolo-[3,4-i]dipyrido[a,c]phenazine (TDP) as electron-acceptor groups are a class of NIR absorbing dyes due to their narrow band gaps, which have been widely studied for NIR-II bioimaging and photothermal therapy (PTT)[15–18]. Consequently, D-A type dyes have the potential to be utilized to achieve NIR-excited PDT, despite they have been rarely studied as photosensitizers for PDT.

It is known that high intersystem crossing (ISC) efficiency of the photosensitizer is favorable to boost $^1O_2$ generation. The commonly used strategy to facilitate the ISC process is to enhance spin-orbit coupling (SOC) of the photosensitizer by introducing heavy atoms or special organic moieties. However, this approach still has some drawbacks and limitations[19,20]. Recently, supramolecular self-assembly has provided an intelligent, "bottom-up" approach to fabricate functional materials at a molecular level[21,22], offering various advantages in phototherapy: (1) The self-organized photosensitizer may have the potential to form band-like electronic structures with much denser energy-level distributions owing to the orbital overlaps between building blocks. Thus, energy matching between the lowest singlet excited state ($S_1$) and the adjacent triplet excited state ($T_n$) will be improved, which favors the ISC process[23,24]. (2) Supramolecular assembly will decrease the non-radiative relaxation based on the restriction of intramolecular motion (RIM) mechanism, which is also beneficial for the ISC process[25,26]. (3) The supramolecular assembly can protect the inside monomeric molecules from oxidation and decomposition, thereby greatly increasing the photostability of the photosensitizer[27]. In view of the above advantages, we believe that the supramolecular photosensitizer system could be a promising alternate to resolve the drawbacks of PDT.

Ru(II)-based complexes are a promising platform in medicine for their potential applications in disease diagnosis and treatment due to their unique and attractive biological properties[28–34]. Furthermore, the rich excited states and tunable photophysicochemical properties of the Ru(II)-based complexes provide great advantage for the development of Ru(II)-based photosensitizers[35–40]. The notable example is a Ru (II) poly-pyridyl complex, TLD-1433, which is currently in a phase II clinical trial as a photosensitizer for treating non-muscle-invasive bladder cancer (NMIBC)[41]. Besides, organometallic Ru(II)-arene complexes have been widely explored as chemotherapeutic agents for cancer treatment owing to their low toxicity and facile modification[42–45]. The ionic character of organometallic Ru(II)-arene complexes can not only improve the poor solubility of the D-A chromophores in common solvents, but also enhance the assembly of the D-A chromophores. Furthermore, the pseudo-octahedral half-sandwich structures of the organometallic Ru(II)-arene complexes may sterically prevent H-aggregation of the D-A type chromophores, thus favoring the formation of J-aggregation with red-shifted absorption bands. Nevertheless, the inherent drawbacks of Ru(II)-arene complexes such as low stability and/or poor bioavailability may compromise the therapeutic efficacy and in vivo performance of Arene-Ru(II) complexes. However, it is demonstrated that encapsulation of ruthenium complexes with biocompatible polymer through physical encapsulation or covalent conjugation has the potential to overcome these drawbacks[46].

In this work, we report a NIR-triggered D-A conjugated Ru(II)-arene complex (RuDA) through the coordination bond between a D-A-D chromophore and the Ru(II)-arene moiety. The resulting complex can self-assemble into metallosupramolecular vesicles in water through noncovalent interactions. Notably, the supramolecular assembly endows RuDA with aggregation-induced intersystem crossing characteristic, thus leading to greatly improved ISC efficiency, which is highly favorable for PDT (Fig. 1A). In order to enhance the tumor accumulation and biocompatibility in vivo, the FDA-approved Pluronic F127 (PEO-PPO-PEO) was used to encapsulate RuDA[47–49], yielding nanoparticles RuDA-NPs (Fig. 1B), which act as a highly efficient PDT/PTT dual-modal agent. In cancer phototherapy (Fig. 1C), RuDA-NPs was applied to treat with MDA-MB-231-tumor bearing nude mice to explore in vivo PDT and PTT efficacy.

## Results and discussion

RuDA consisting of TPA and TDP functional groups (Fig. 2A) was prepared following the procedure shown in Supplementary Fig. 1, RuDA was characterized by $^1H$ and $^{13}C$ NMR spectroscopy, electrospray ionization mass spectrometry along with elemental analysis (Supplementary Figs. 2–4). The map of the electron density difference of RuDA for the lowest singlet transition was calculated to investigate the charge transfer process by the time-dependent density functional theory (TD-DFT) method. As shown in Supplementary Fig. 5, the electron density mainly drifts from the triphenylamine to the acceptor unit TDP after photoexcitation, which can be assigned to the typical intramolecular charge transfer (CT) transition.

The self-assembly property of RuDA was investigated in mixtures of DMF and water with different ratios by UV–vis absorption spectroscopy. As shown in Fig. 2B, RuDA exhibits an absorption band at 600–900 nm with a maximum at 729 nm in DMF. Increase of the water fraction results in a gradual red-shift of the maximum absorption peak of RuDA to 800 nm, indicating the J-aggregation of RuDA in the assembled system. The photoluminescence spectra of RuDA in different solvents are shown in Supplementary Fig. 6. Obviously, RuDA exhibits a typical NIR-II luminescence with maximum emission wavelength at ca. 1050 nm in $CH_2Cl_2$ and $CH_3OH$, respectively. The large stokes shift of RuDA (around 300 nm) indicates the considerable change in excited state geometry and the formation of low-energy excited state. The luminescent quantum yields of RuDA are determined to be 3.3% and 0.6% in $CH_2Cl_2$ and $CH_3OH$, respectively. However, a slight red-shift in emission and decreased quantum

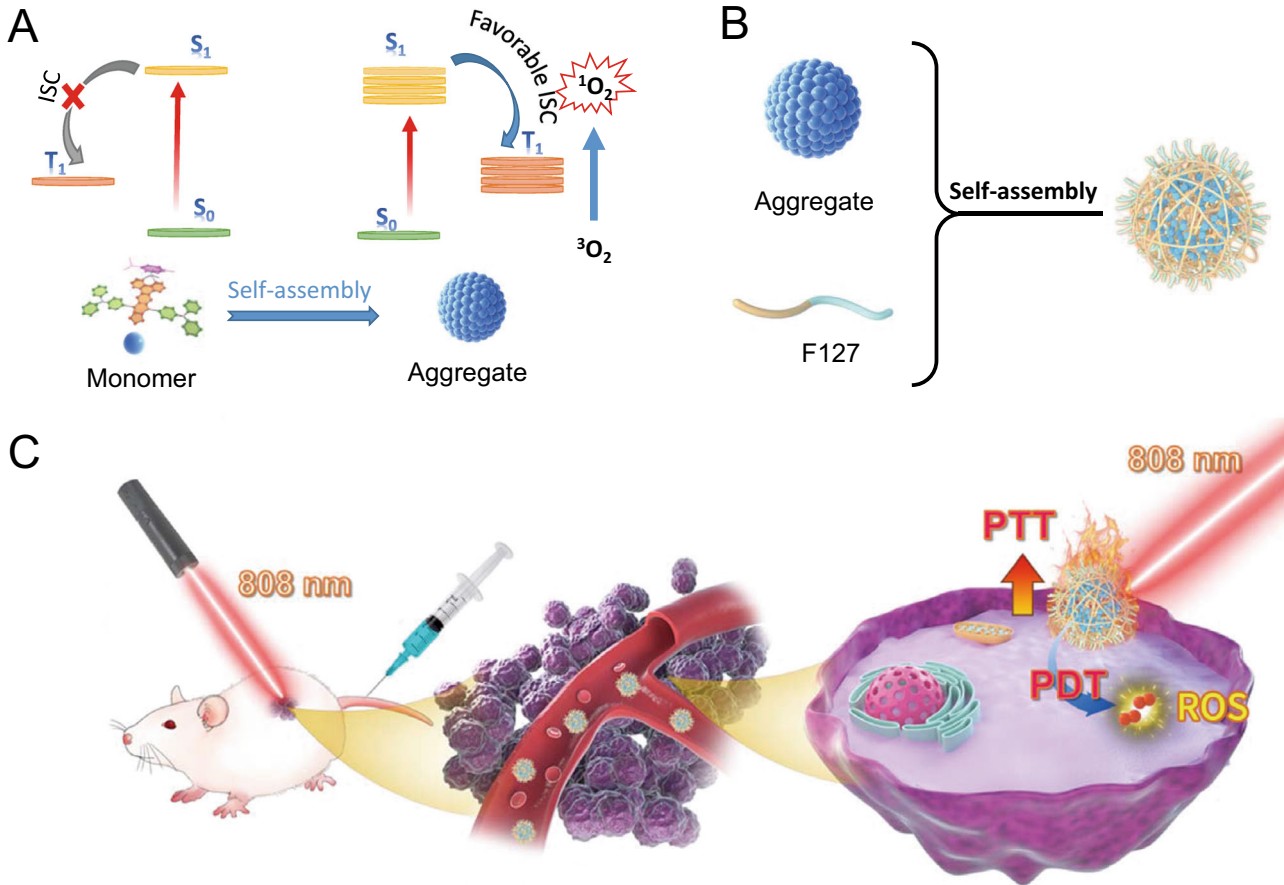

**Fig. 1 Schematic illustration of RuDA-NPs for phototherapy. A** Schematic illustration of the photophysical mechanism of RuDA in monomer and aggregate forms for cancer phototherapy, **B** synthesis of RuDA-NPs, and **C** RuDA-NPs for NIR-activated PDT and PTT.

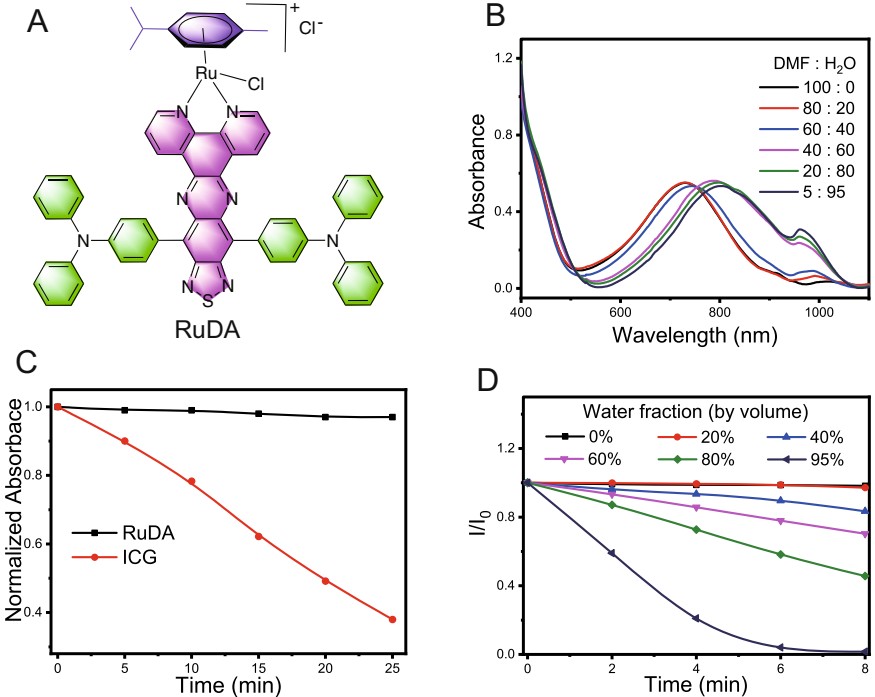

**Fig. 2 Photo-physicochemical properties of RuDA. A** Chemical structure of RuDA. **B** Absorption spectra of RuDA in mixtures of DMF and water at various ratios. **C** Time-dependent variation in the normalized absorbance values of RuDA (800 nm) and ICG (779 nm) under 808 nm laser irradiation at 0.5 W cm$^{-2}$. **D** Photodegradation of ABDA indicative of $^1O_2$ generation induced by RuDA in DMF/H$_2$O mixtures with different water fractions under 808 nm laser irradiation at 0.5 W cm$^{-2}$.

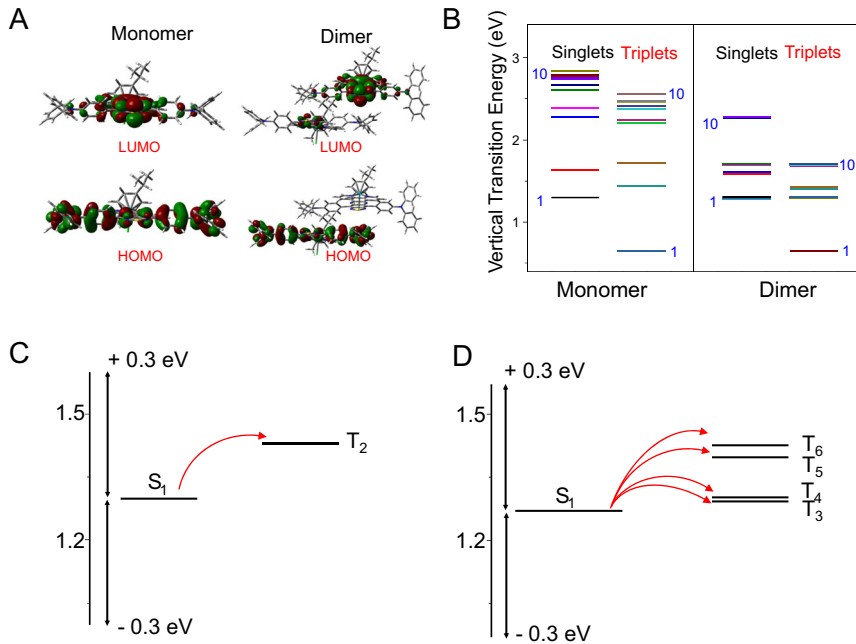

**Fig. 3 Quantum-chemical calculations of RuDA. A** Calculated HOMOs and LUMOs of RuDA in monomeric and dimeric forms. **B** Singlet and triplet energy levels of of RuDA in monomer and dimer, respectively. **C** Calculated energy levels and possible ISC channels of RuDA in monomeric **C** and dimeric **D** forms. The arrows refer to the possible ISC channels.

yield (0.22%) are observed in a mixture of methanol and water (5/95, v/v), probably attributing to the self-assembly of RuDA.

To intuitively observe the self-assembly of RuDA, we applied liquid atomic force microscopy (AFM) to image the morphology change upon addition of water to methanol solution of RuDA. When the water fraction is below 80%, no obvious aggregation is observed (Supplementary Fig. 7). However, upon a further increment of water fraction to 90–95%, small nanoparticles appear, demonstrating that the self-assembly of RuDA occurs. In addition, irradiation with 808 nm laser does not have an influence on the absorbance intensity of RuDA in aqueous solution (Fig. 2C and Supplementary Fig. 8). On the contrary, the absorbance of indocyanine green (ICG, as a control) decreases fast at 779 nm, suggesting the excellent photostability of RuDA. Furthermore, the stability of RuDA-NPs was investigated in the media of PBS (pH = 5.4, 7.4, and 9.0), 10% FBS, and DMEM (high glucose) using UV-vis absorption spectra at different time points. As exhibited in Supplementary Fig. 9, negligible changes of the absorption bands of RuDA-NPs were observed in PBS at pH 7.4/9.0, FBS, and DMEM, indicating the excellent stability of RuDA-NPs. However, hydrolysis of RuDA was detected under acidic condition (pH = 5.4). We also use high-performance liquid chromatography (HPLC) technique to evaluate the stability of RuDA and RuDA-NPs further. As exhibited in Supplementary Fig. 10, RuDA is stable in the first hour in the mixture of methanol and water (50/50, v/v), and the hydrolysis was observed after 4 h. However, only a broad bump peak was observed for RuDA-NPs. Hence, gel permeation chromatography (GPC) was used to evaluate the stability of RuDA-NPs in the medium of PBS (pH = 7.4). As shown in Supplementary Fig. 11, negligible changes of the peak height, peak width and peak area of RuDA-NPs were observed after 8 h incubation under the test conditions, indicating the excellent stability of RuDA-NPs. In addition, TEM images demonstrated that the morphology of RuDA-NPs nanoparticles almost remained unchanged after 24 h in diluted PBS buffer (pH = 7.4, Supplementary Fig. 12).

Since the self-assembly could endow RuDA with different functionalities and chemical features, the photochemical properties of RuDA were investigated by observing the degradation of 9,10-anthracenediylbis(methylene)-dimalonic acid (ABDA, a $^1O_2$ indicator) in methanol-water mixtures with different water fractions[50]. As shown in Fig. 2D and Supplementary Fig. 13, no ABDA degradation is observed when water fraction is below 20%. As the water fraction is increased to 40%, the degradation of ABDA occurs as evidenced by the decrease of fluorescence intensity of ABDA. It is also observed that higher water content results in faster the degradation, indicating that the self-assembly of RuDA is essential and beneficial for ABDA degradation. This phenomenon is totally different from that of current ACQ (aggregation-caused quenching) chromophores. The $^1O_2$ quantum yield of RuDA is calculated to be 16.4% in 98% $H_2O$/2% DMF upon 808 nm laser irradiation, which is 82-fold higher than that of ICG ($\Phi_\Delta$ = 0.2%)[51], demonstrating the significant $^1O_2$ generation efficiency of RuDA in the aggregation state.

Electron spin resonance (ESR) spectroscopy with 2,2,6,6-tetramethyl-4-piperidone (TEMP) and 5,5-dimethyl-1-pyrroline N-oxide (DMPO) as the spin-trapping agents was utilized to identify the ROS species generated by RuDA. As shown in Supplementary Fig. 14, the enhanced triplet ESR signal of 2,2,6,6-tetramethyl-4-piperidone-N-oxyl (TEMPO) generated by the reaction of TEMP with $^1O_2$ was observed as the irradiation time increased from 0 to 4 min, confirming the generation of $^1O_2$. Besides, the typical 1:2:2:1 four-line ESR signal of DMPO-OH· adducts was detected when RuDA was incubated with DMPO upon irradiation, demonstrating the generation of hydroxyl radical (OH·). Overall, the above results indicate the ability of RuDA to promote the ROS generation via a dual type I/II photosensitization process.

For better understanding the electronic characters of RuDA in monomer and aggregate forms, frontier molecular orbitals of RuDA in monomeric and dimeric forms were calculated by DFT method. As shown in Fig. 3A, the highest occupied molecular orbital (HOMO) of RuDA in monomer state is delocalized along the ligand backbone, while the lowest unoccupied molecular orbital (LUMO) is centered on the acceptor unit TDP. In

contrast, the electron density in the HOMO of the dimer is localized on the ligand of one RuDA molecule, while that in the LUMO is mainly localized on the acceptor unit of the other RuDA molecule, suggesting the potential intermolecular CT character of RuDA in dimer.

The distribution of electrons and holes for the low-energy singlet excited states of RuDA in monomeric and dimeric forms were analyzed using Multiwfn 3.8 software[52,53], in which calculations were carried out using the TD-DFT approach. As shown in Supplementary Tabs. 1–2, holes of monomeric RuDA are mainly delocalized along the ligand backbone in these singlet excited states, while electrons are mainly located at the TDP group, showing the intramolecular CT characteristics. Besides, more or less overlaps of holes and electrons were observed for these singlet excited states, indicating that these singlet excited states have some contributions from the local excitation (LE). As for the dimer, in addition to the intramolecular CT and LE character, a proportion of intermolecular CT character was observed in the relevant states, especially for $S_3$, $S_4$, $S_7$ and $S_8$, which are dominated by intermolecular CT transition based on intermolecular CT analysis (Supplementary Tab. 3).

To better understand the experimental results, we further investigated the properties of the excited states of RuDA to probe the difference between the monomer and dimer (Supplementary Tabs. 4–5). As shown in Fig. 3B, the energy levels of the singlet and triplet excited states of the dimer are much denser than those of the monomer, which is beneficial for decreasing the energy gap between $S_1$ and $T_n$. It has been reported that the ISC transitions could be realized within small energy gap ($\Delta E_{S1-Tn} < 0.3$ eV) between $S_1$ and $T_n$[54]. Besides, only one orbital, either the occupied or the unoccupied, should be different in the relevant singlet and triplet states to ensure non-zero SOC integrals[55]. Thus, based on excitation energies and orbital transition analysis, all the possible ISC transition channels are given in Fig. 3C, D. Notably, only one ISC channel is available in the monomer, while four ISC channels are accessible in the dimeric form, which could enhance the ISC transition. Therefore, it is reasonable to infer that the more RuDA molecules aggregate, the more ISC channels are accessible. Consequently, the aggregates of RuDA can form two band-like electronic structures in the singlet and triplet states with decreased the energy gap between $S_1$ and available $T_n$, which promoting the ISC efficiency to facilitate $^1O_2$ production.

To further elucidate the underlying mechanism, we synthesized a contrast compound of arene-Ru(II) complex (RuET) by replacing two phenyl groups of triphenylamine in RuDA with two ethyl groups (Fig. 4A, for full characterization, see ESI, Supplementary Figs. 15–21). RuET has the same intramolecular CT characteristics as RuDA from the donor (diethylamine) to the acceptor (TDP). As expected, the absorption spectra of RuET in DMF display a low-energy charge transfer band with an intense NIR absorption in 600–1100 nm regions (Fig. 4B). Moreover, the aggregation of RuET is also observed with increasing the water fraction, as reflected by the red-shift of the absorption maxima, which is further confirmed by the liquid AFM imaging (Supplementary Fig. 22). The results show that RuET, similar to RuDA, can form the intramolecular state and self-assemble into aggregate structures.

The photo-degradation of ABDA in the presence of RuET was evaluated under 808 nm laser irradiation. Surprisingly, no degradation of ABDA is observed in the different water fractions (Supplementary Fig. 23). One possible reason is that RuET cannot effectively form the band-like electronic structures because the ethyl chain is unfavorable for highly efficient intermolecular charge transfer. Therefore, electrochemical impedance spectroscopy (EIS) and transient photocurrent measurements were carried out to compare the photoelectrochemical property of RuDA

with RuET. According to the Nyquist plots (Fig. 4C), RuDA shows a much smaller radius than RuET, hinting the faster intermolecular electron transport and better conductivity of RuDA[56]. Moreover, the photocurrent density of RuDA is much higher than that of RuET (Fig. 4D), confirming the better charge transport efficiency of RuDA[57]. Accordingly, the phenyl group of triphenylamine in RuDA plays an important role in facilitating the intermolecular charge transfer and forming the band-like electronic structures.

To enhance the tumor accumulation and biocompatibility in vivo, we further encapsulated RuDA with F127. The average hydrodynamic diameter of RuDA-NPs was determined as 123.1 nm with a narrow distribution (PDI = 0.089) by using the dynamic light scattering (DLS) technique (Fig. 5A), which is favorable for tumor accumulation via enhanced permeability and retention (EPR) effect. TEM image shows that RuDA-NPs have a uniform spherical shape with an average diameter of 86 nm. Significantly, the maximum absorption peak of RuDA-NPs appears at 800 nm (Supplementary Fig. 24), indicating that RuDA-NPs may maintain the functions and properties of the self-assembled RuDA. The ROS quantum yield of RuDA-NPs was calculated to be 15.9%, which is comparable to RuDA. The photothermal properties of RuDA-NPs were studied using an infrared camera under 808 nm laser irradiation. As depicted in Fig. 5B, C, negligible temperature increase is observed for the control group (PBS only), whereas the temperatures of the RuDA-NPs solution are rapidly elevated with temperature increases ($\Delta T$) of 15.5, 26.1, and 43.0 °C at the concentrations of 25, 50, and 100 µM, respectively, suggesting the strong photothermal effect of RuDA-NPs. In addition, heating/cooling cycle measurement was carried out to evaluate the photothermal stability of RuDA-NPs, along with ICG for comparison. There is no decay in temperature for RuDA-NPs after five heating/cooling cycles (Fig. 5D), indicating the excellent photothermal stability of RuDA-NPs. On the contrary, ICG displays less photothermal stability as revealed by the obvious decay in the photothermal temperature plateau under the same conditions. According to the previous method[58], the photothermal conversion efficiency (PCE) of RuDA-NPs was calculated to be 24.2%, which is higher than those of existing photothermal materials, such as gold nanorods (21.0%) and gold nanoshells (13.0%)[59]. As a consequence, RuDA-NPs display excellent photothermal performance, which makes it a promising PTT agent.

The in vitro photocytotoxicity of RuDA-NPs was evaluated against MDA-MB-231 human breast cancer cells. As shown in Fig. 6A, B, both RuDA-NPs and RuDA exhibit negligible cytotoxicity without irradiation, implying the low dark toxicity of RuDA-NPs and RuDA. However, once exposed to 808 nm laser irradiation, RuDA-NPs and RuDA show severe photocytotoxicity against MDA-MB-231 cancer cells with $IC_{50}$ values (half maximal inhibitory concentrations) of 5.4 and 9.4 µM, respectively, demonstrating that RuDA-NPs and RuDA have the potential in cancer phototherapy. Besides, the photocytotoxicity of RuDA-NPs and RuDA was further investigated in the presence of ROS-scavenger Vitamin C (Vc) to elucidate the role of ROS in photo-induced cytotoxicity. Distinctly, the cell viability increases upon addition of Vc with $IC_{50}$ values of 25.7 and 40.0 µM for RuDA-NPs and RuDA, respectively, demonstrating the essential role of ROS in the photocytotoxicity of RuDA-NPs and RuDA. The photo-induced cytotoxicity of RuDA-NPs and RuDA was further confirmed in MDA-MB-231 cancer cells by the live/dead cell staining assay with calcein AM (green fluorescence for living cells) and propidium iodide (PI, red fluorescence for dead cells) as fluorescence probes. As depicted in Fig. 6C, the cells treated with RuDA-NPs or RuDA remain alive without irradiation as visualized by the intense green fluorescence. On the contrary, only red

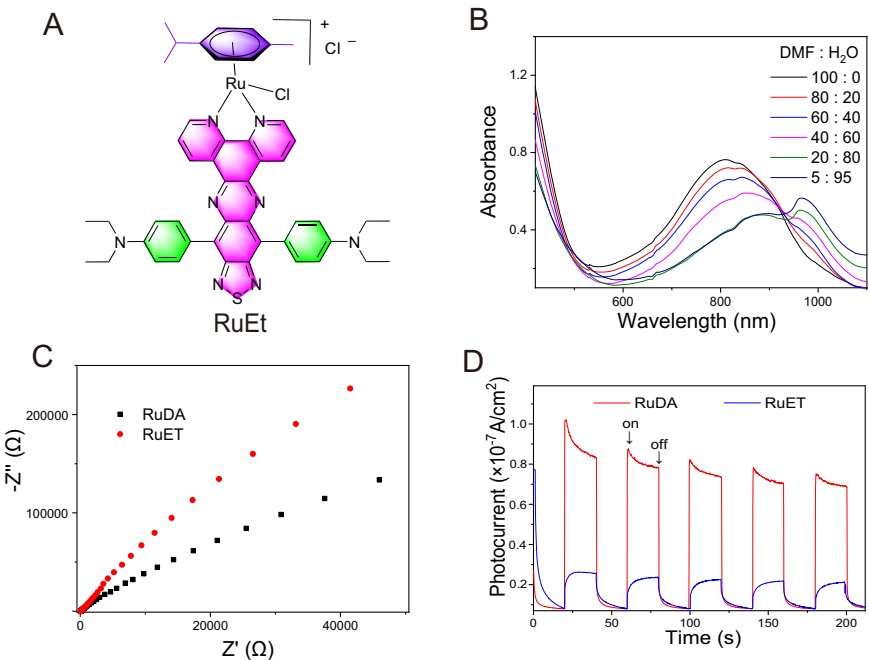

**Fig. 4 Photoelectrochemical properties of RuDA and RuET. A** Chemical structure of RuET. **B** Absorption spectra of RuET in mixtures of DMF and water at various ratios. **C** EIS Nyquist plots of RuDA and RuET. **D** Photocurrent responses of RuDA and RuET under 808 nm laser irradiation.

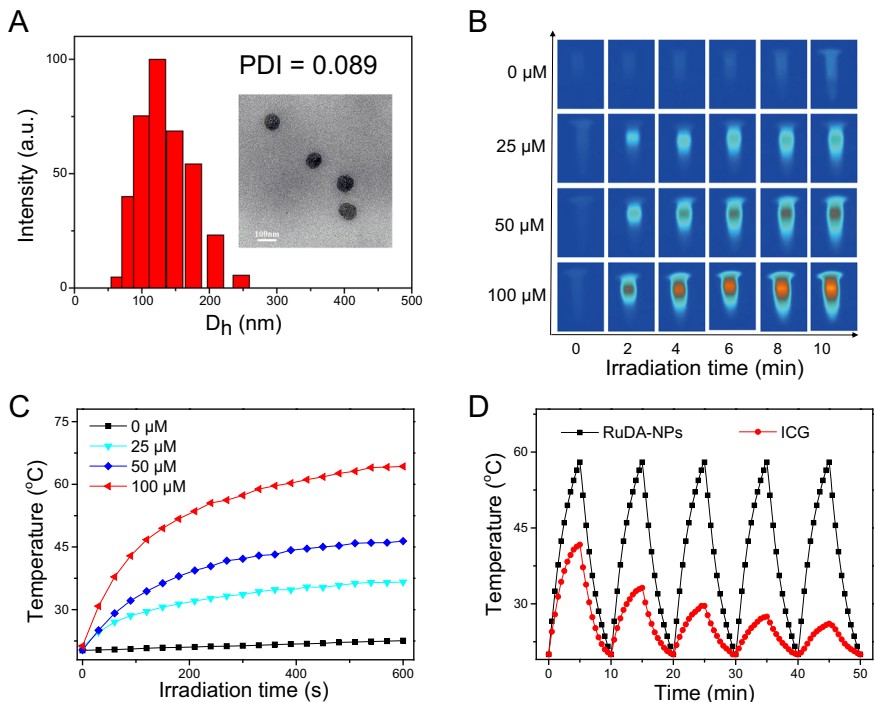

**Fig. 5 The characterization and photothermal properties of RuDA-NPs. A** DLS analysis and TEM image (inset) of RuDA-NPs. **B** Thermal images of RuDA-NPs at different concentrations under 808 nm (0.5 W cm$^{-2}$) laser irradiation. **C** Photothermal conversion profile of RuDA-NPs at different concentrations, which is the quantitative data of **B**. **D** Temperature elevation of RuDA-NPs and ICG during five cycles of heating-cooling processes.

fluorescence is observed upon laser irradiation, confirming the efficient photocytotoxicity of RuDA-NPs or RuDA. Notably, the green fluorescence appears upon the addition of Vc, suggesting the compromised photocytotoxicity of RuDA-NPs and RuDA. These results are in accordance with the in vitro photocytotoxicity assay.

The intracellular ROS generation in MDA-MB-231 cells was investigated using a 2,7-dichlorodihydrofluorescein diacetate (DCFH-DA) staining method. As shown in Fig. 6D, the cells treated with RuDA-NPs or RuDA display obvious green fluorescence under 808 nm laser irradiation, indicating the effective ROS generation abilities of RuDA-NPs and RuDA. On the contrary, only weak fluorescence signals are observed for cells in the absence of light or in the presence of Vc, which is indicative of the negligible ROS production. The intracellular ROS levels in RuDA-NPs- and RuDA-treated MDA-MB-231 cells were further

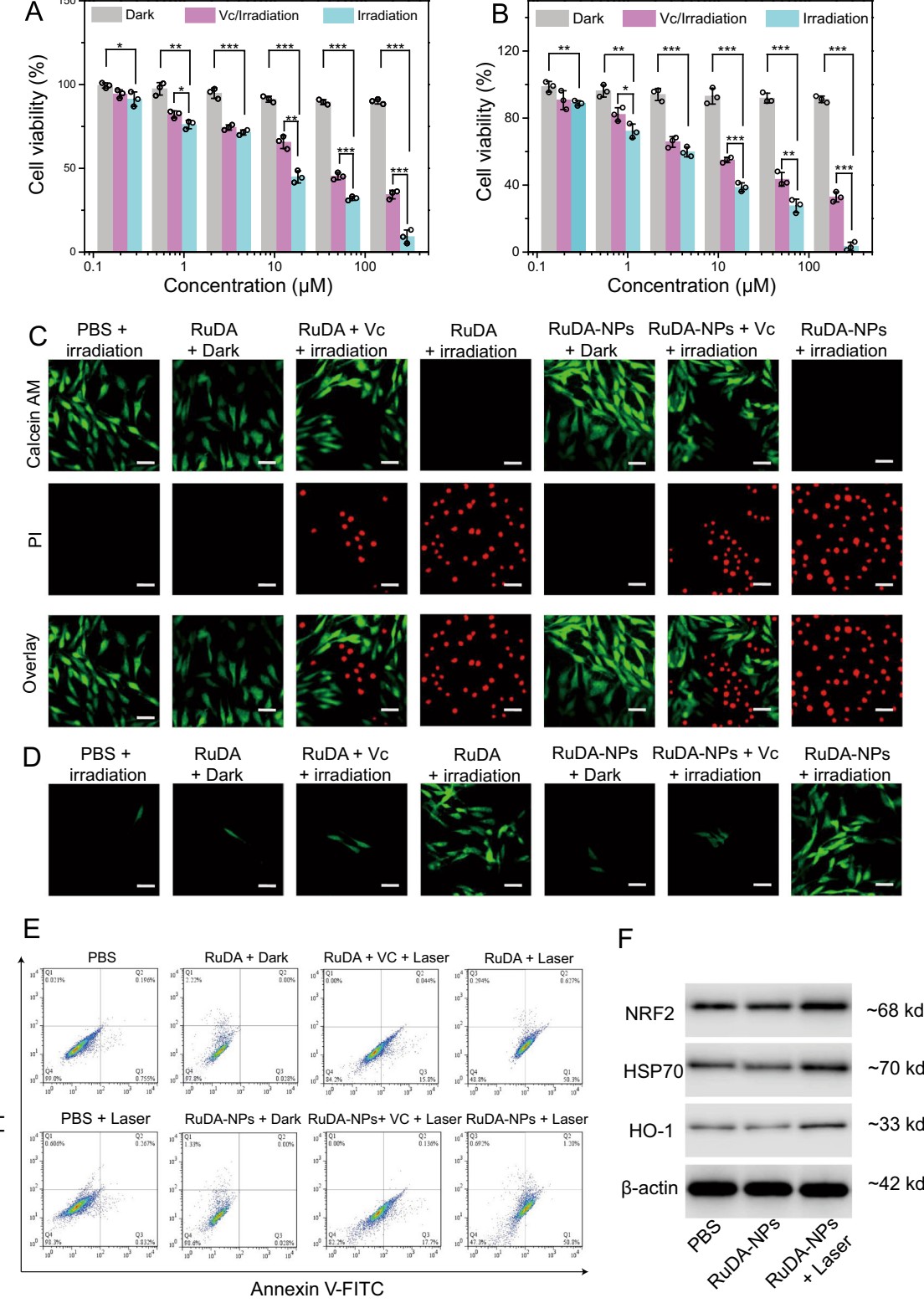

determined by flow cytometry. As shown in Supplementary Fig. 25, both RuDA-NPs and RuDA under 808 nm laser irradiation produce a significant increase of the mean fluorescence intensity (MFI) by approximate 5.1- and 4.8-fold as much as control group, respectively, confirming their excellent ROS generation abilities. However, the intracellular ROS level in either RuDA-NPs- or RuDA-treated MDA-MB-231 cells is only comparable to the control group without laser irradiation or in the presence of Vc, similar to the result of confocal fluorescence analysis.

It has been proved that mitochondrion is a main target of Ru(II)-arene complexes[60]. Thus, the subcellular localizations of RuDA and RuDA-NPs were investigated. As shown in Supplementary Fig. 26, RuDA and RuDA-NPs exhibit a similar cellular distribution profile, with the highest accumulation in mitochondria ($62.5 \pm 4.3$ and $60.4 \pm 3.6$ ng/mg protein, respectively).

**Fig. 6 In vitro phototherapy efficiencies of RuDA and RuDA-NPs. A** RuDA- and **B** RuDA-NPs-dose dependent cell viabilities in MDA-MB-231 cells in the presence or absence of Vc (0.5 mM), respectively. Error bars, mean ± SD ($n = 3$). Unpaired, two-sided t tests *$p < 0.05$, **$p < 0.01$, and ***$p < 0.001$. **C** Live/dead cell staining assays using calcein AM and propidium iodide as fluorescence probes. Scale bars: 30μm. A representative image of three biological replicates from each group is shown. **D** Confocal fluorescence images of ROS generation in MDA-MB-231 cells under different treatment conditions. Green fluorescence from DCF indicates the presence of ROS. Irradiation was performed using 808 nm laser at 0.5 W cm$^{-2}$ for 10 min (300 J cm$^{-2}$). Scale bars: 30 μm. A representative image of three biological replicates from each group is shown. **E** Flow cytometry analysis for apoptosis of MDA-MB-231 cells treated with RuDA-NPs (50 μM) or RuDA (50 μM) in the presence and absence of Vc (0.5 mM), and irradiated with or without 808 nm laser (0.5 W cm$^{-2}$) for 10 min. A representative image of three biological replicates from each group is shown. **F** Cellular Nrf-2, HSP70, and HO-1 expressions of MDA-MB-231 cells treated with RuDA-NPs (50 μM) with or without 808 nm laser irradiation (0.5 W cm$^{-2}$, 10 min, 300 J cm$^{-2}$). A representative image of two biological replicates from each group is shown.

However, only a small amount of ruthenium is observed in the nuclear fraction for RuDA and RuDA-NPs (3.5% and 2.1%, respectively). The residual cellular fraction contains the remaining of the ruthenium, 31.7% (30.6 ± 3.4 ng/mg protein) for RuDA and 42.9% (47.2 ± 4.5 ng/mg protein) for RuDA-NPs, respectively. Overall, RuDA and RuDA-NPs are mainly accumulated in the mitochondria. To assess the mitochondrial dysfunction, we applied JC-1 and MitoSOX Red staining to evaluate the mitochondrial membrane potential and superoxide generation capacities, respectively. As shown in Supplementary Fig. 27, intense green (JC-1) and red (MitoSOX Red) fluorescence is observed in both the RuDA and RuDA-NPs-treated cells under 808 nm laser irradiation, demonstrating that both RuDA and RuDA-NPs can induce mitochondrial membrane depolarization and superoxide generation effectively. Additionally, the cell death mechanism was determined using flow cytometry based on annexin V-FITC/propidium iodide (PI) assay. As depicted in Fig. 6E, upon 808 nm laser irradiation, both RuDA and RuDA-NPs induce a significantly increased incidence of early stage apoptosis (lower right quadrant) in MDA-MB-231 cells compared with the PBS- or PBS plus laser-treated cells. However, when Vc is added, the apoptotic rates of RuDA and RuDA-NPs are significantly reduced to 15.8% and 17.8% from 50.9% and 52.0%, respectively, confirming the essential role of ROS in the photocytotoxicity of RuDA and RuDA-NPs. Besides, negligible necrotic cells (upper left quadrant) are observed for all the test groups, demonstrating that apoptosis might be a main form of the RuDA- and RuDA-NPs-induced cell death.

As oxidative stress injury is a major determinant of cell apoptosis[61], the expression of the nuclear factor erythroid 2-related factor 2 (Nrf2)[62], a key regulator of the antioxidant system, was investigated in RuDA-NPs-treated MDA-MB-231 cells to elucidate the modes of action induced by RuDA-NPs under irradiation. Meanwhile, the expression of its downstream protein, heme oxygenase-1 (HO-1), was also assayed. As shown in Fig. 6F and Supplementary Fig. 29, RuDA-NPs-mediated phototherapy upregulates the expression levels of Nrf2 and HO-1 as compared with the PBS group, demonstrating that RuDA-NPs can stimulate the oxidative stress signaling pathway. Besides, the expression of the heat-responsive heat shock protein Hsp70 was evaluated to explore the photothermal effect of RuDA-NPs[63]. Obviously, the cells treated with RuDA-NPs + 808 nm laser irradiation exhibit an increased Hsp70 expression compared with the other two groups, reflecting the cellular responses to hyperthermia.

The prominent in vitro results promote us to study the in vivo profile of RuDA-NPs in MDA-MB-231-tumor bearing nude mice. The tissue distribution of RuDA-NPs was investigated by characterizing the ruthenium content in the liver, heart, spleen, kidney, lung, and tumor. As depicted in Fig. 7A, the maximum contents of RuDA-NPs in the normal organs are found at the first observation time point (4 h), whereas maximum content is detected at 8 h post-injection in the tumor tissues, possibly due to

the EPR effect of RuDA-NPs. Upon the distribution results, 8 h post-injection was adopted as the optimum treatment time for RuDA-NPs. In order to illustrate the process of RuDA-NPs accumulation at the tumor site, the photoacoustic (PA) performance of RuDA-NPs was monitored by recording PA signals of RuDA-NPs at different post-injection times. First, the in vivo PA signals of RuDA-NPs was evaluated by recording the PA images of tumor sites after intratumoral injection of the RuDA-NPs. As shown in Supplementary Fig. 30, RuDA-NPs display strong PA signals, and there is a positive correlation between the concentrations of the RuDA-NPs and the intensity of the PA signals (Supplementary Fig. 30A). Then the in vivo PA images of tumor sites after intravenous injection of RuDA and RuDA-NPs were recorded at varied time post-injection points. As shown in Fig. 7B, the PA signals of RuDA-NPs from the tumor sites gradually enhance with time, and reach a plateau at 8 h of post injection, which is in accordance with the tissue distribution results determined by ICP-MS analysis. As for RuDA (Supplementary Fig. 30B), the maximum PA signal intensity appears at 4 h of post injection, implying a rapid tumor penetration rate of RuDA. Additionally, the excretion behaviors of RuDA and RuDA-NPs were explored by detecting the ruthenium amount in urine and feces using ICP-MS. The primary excretion pathway of RuDA (Supplementary Fig. 31) and RuDA-NPs (Fig. 7C) is through feces, and efficient clearance of RuDA and RuDA-NPs is observed within the study period of 8 d, implying that RuDA and RuDA-NPs could be effectively excreted from the body without long-term toxicity.

The in vivo heat generation capacities of RuDA-NPs were studied in MDA-MB-231-tumor bearing nude mice with RuDA for comparison. As shown in Fig. 8A and Supplementary Fig. 32, the control (saline) group exhibits small temperature variation ($\Delta T \approx 3\,^\circ C$) after 10 min continuous irradiation. However, rapid temperature elevations occur for RuDA-NPs and RuDA with maximum temperatures of 55.2 and 49.9 °C, respectively, enabling adequate hyperthermia for in vivo cancer therapy. Compared with RuDA ($\Delta T \approx 19\,^\circ C$), a high temperature increase ($\Delta T \approx 24\,^\circ C$) is observed for RuDA-NPs possibly attributing to its better permeability and accumulation in the tumor tissues via EPR effect.

The in vivo phototherapeutic efficacy of RuDA-NPs and RuDA was evaluated, in which the nude mice bearing MDA-MB-231 tumors were intravenously administrated with RuDA-NPs or RuDA through the tail vein at a single dose of 10.0 μmol kg$^{-1}$, followed by 808 nm laser irradiation at 8 h post-injection. As shown in Fig. 8B, the tumor volumes increase significantly in the saline and laser-treated groups, indicating that saline or 808 laser exposure has negligible effect on tumor growth. Similar to that in the saline group, rapid tumor growth is also observed in the mice treated with RuDA-NPs or RuDA in the absence of laser irradiation, exhibiting their low dark toxicity. In contrast, after laser irradiation, both RuDA-NPs and RuDA treatments induce significant tumor regression with 95.2% and 84.3% reduction in

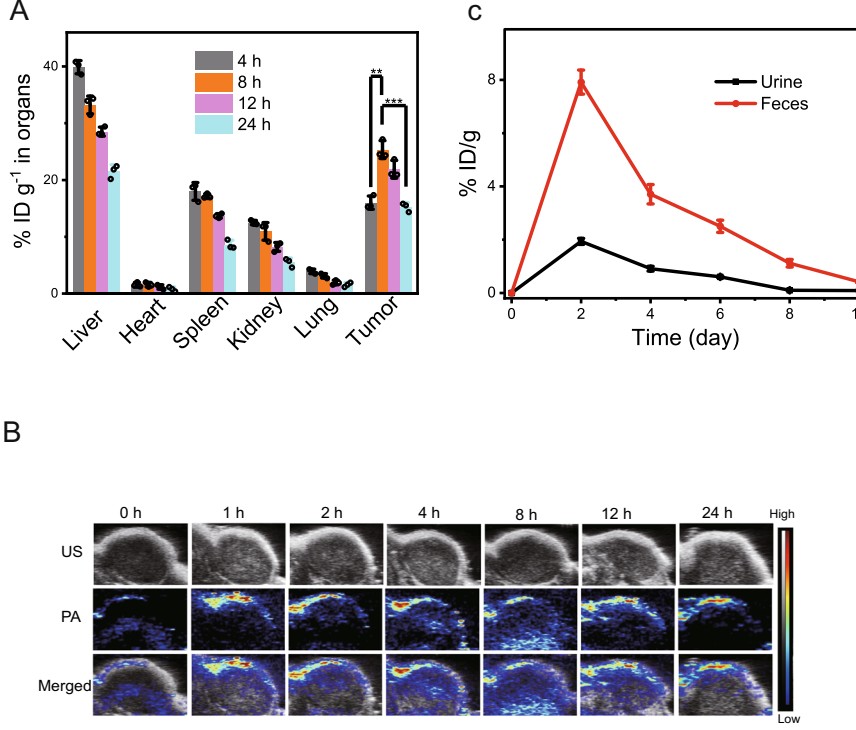

**Fig. 7 Ex vivo biodistribution and in vivo PA imaging. A** Ex vivo tissue distribution of RuDA-NPs in mice determined by the content of Ru (% injected dose (ID) of Ru per gram of tissues) at different post injection time. The data represent the mean ± SD (n = 3). Unpaired, two-sided t tests *p < 0.05, **p < 0.01, and ***p < 0.001. **B** In vivo PA images of tumor sites under excitation at 808 nm after intravenous injection of RuDA-NPs (10 μmol kg⁻¹) at different time points. **C** Ru excreted out of the mice body via urine and feces after intravenous administration of RuDA-NPs (10 μmol kg⁻¹) at different time intervals. The data represent the mean ± SD (n = 3).

tumor volume, respectively, as compared with the saline treated group, indicating the excellent synergistic PDT/PTT effect mediated by RuDA-NPs or RuDA. In comparison with RuDA, RuDA-NPs exhibit much preferable phototherapeutic efficacy, which is mainly owing to the EPR effect of RuDA-NPs. The tumor growth inhibition results were further evaluated by the tumor weights excised on the 15th day of the treatment (Fig. 8C and Supplementary Fig. 33). The average tumor weights of the RuDA-NPs and RuDA treated mice with irradiation are 0.08 and 0.27 g, respectively, which are much lighter than that of the control group (1.43 g).

Besides, the body weights of the mice were recorded every three days to investigate the in vivo dark toxicity of RuDA-NPs or RuDA. As shown in Fig. 8D, no obvious difference in body weights is observed for all the treated groups. Furthermore, the hematoxylin and eosin (H&E) staining of the major organs (heart, liver, spleen, lung, and kidney) from different treatment groups were undertaken. As shown in Fig. 8E, the H&E staining images of five major organs from the RuDA-NPs and RuDA groups exhibit no obvious abnormalities or organ damages. These results demonstrate that both RuDA-NPs and RuDA have no sign of in vivo toxicity. Moreover, H&E staining images of tumors showed that both the RuDA + Laser and RuDA-NPs + Laser groups could cause severe cancer cell destruction, demonstrating the excellent in vivo phototherapeutic efficacy of RuDA and RuDA-NPs.

In summary, an organometallic Ru(II)-arene complex (RuDA) bearing a D-A type ligand has been designed to take advantage of the aggregation approach to facilitate the ISC process. The synthesized RuDA can self-assemble to form RuDA-derived supramolecular systems via non-covalent interactions, which boosts the ¹O₂ generation alongside with highly efficient photothermal conversion for photoinduced cancer therapy. It is noteworthy that the monomeric RuDA exhibits no $^1O_2$ generation under 808 nm laser irradiation, whereas significant $^1O_2$ production can be generated in its aggregation state, proving the rationality and effectiveness of our design. Subsequent studies show that supramolecular assembly endows the RuDA with improved photophysical and photochemical performances, such as red-shifted absorbance and resistance to photobleaching, which are highly desired for PDT and PTT treatment. Both in vitro and in vivo experiments indicate that RuDA-NPs with good biocompatibility and preferable tumor accumulation have excellent photo-induced anticancer activity under 808 nm laser irradiation. In all, RuDA-NPs, as an efficient PDT/PTT dual-modal supramolecular agent, will enrich the toolbox of photosensitizers activated above the wavelength 800 nm. The proof-of-concept design of supramolecular systems provides an effective approach toward NIR-activated photosensitizers with an excellent photosensitizing effect.

## Methods

**Materials and instrumentation.** All chemicals and solvents were obtained from commercial suppliers, and used without further purification. RuCl₃ was purchased from Boren Precious Metal Co. Ltd (Kunming, China). [(η⁶-p-cym)Ru(phendio)Cl]Cl (phendio = 1,10- phenanthroline-5,6-dione) and 4,7-bis[4-(N,N-diphenyla-mino)phenyl]-5,6-diamino-2,1,3-benzothiadiazole were synthesized according to previous paper[64,65]. NMR spectra were recorded on Bruker Avance III-HD 600 MHz spectrometer in the Analysis and Testing Center of Southeast University, using d⁶-DMSO or CDCl₃ as solvent. Chemical shifts δ are reported in ppm, expressed relative to tetramethylsilane, and coupling constants J are given as absolute values in Hz. High-resolution mass spectrometry (HRMS) was conducted on an Agilent 6224 ESI/TOF MS instrument. Elemental analysis of C, H, and N was carried out on a Vario MICROCHNOS elemental analyzer (Elementar). UV–vis spectra were measured on a Shimadzu UV3600 spectrophotometer. Fluorescence spectra were recorded by a Shimadzu RF-6000 spectrofluorometer. ESR spectra were recorded on a Bruker EMXmicro-6/1 instrument. Morphologies and structures of the prepared samples were examined by FEI Tecnai G20 (TEM) operating at 200 kV and Bruker Icon (AFM). Dynamic light scattering (DLS) was performed

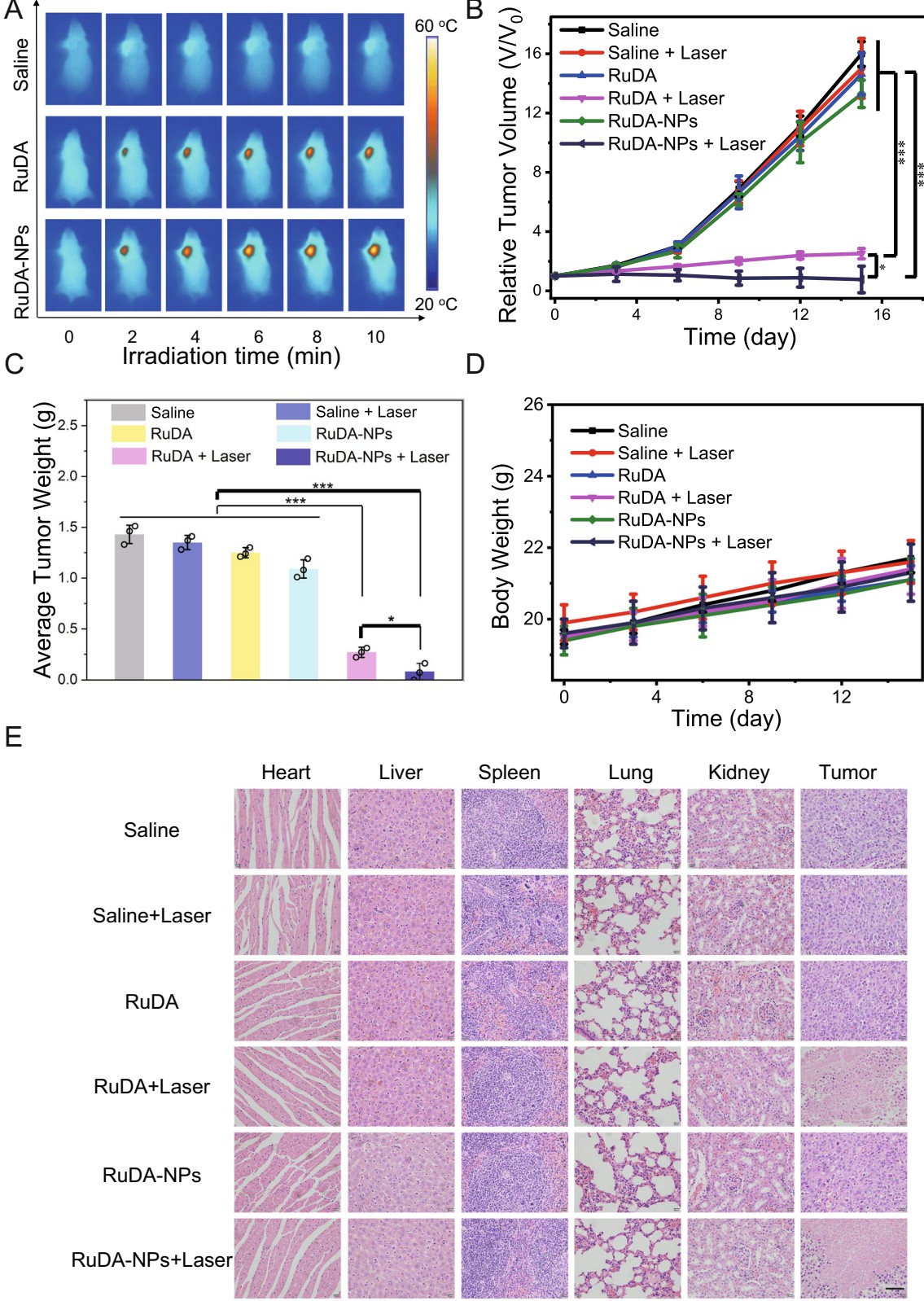

**Fig. 8 In vivo phototherapy efficacies of RuDA and RuDA-NPs on tumors. A** IR thermal images of MDA-MB-231 tumor-bearing mice irradiated with 808 nm laser for different time at 8 h post injection. A representative image of four biological replicates from each group is shown. **B** Relative tumor volume and **C** average tumor weights for different groups of mice during the therapeutic process. **D** Body weight curves of different groups of mice. Irradiation was performed with 808 nm laser at 0.5 W cm$^{-2}$ for 10 min (300 J cm$^{-2}$). Error bars, mean ± SD ($n = 3$). Unpaired, two-sided t tests *$p < 0.05$, **$p < 0.01$, and ***$p < 0.001$. **E** H&E staining images of major organs and tumors from different treatment groups, including Saline, Saline + Laser, RuDA, RuDA + Laser, RuDA-NPs, and RuDA-NPs + Laser groups. Scale bars: 60 μm.

on a Nanobrook Omni analyzer (Brookhaven). The photoelectrochemical properties were measured on an electrochemical system (CHI-660, China). Photoacoustic images were obtained using a FUJIFILM VisualSonics Vevo® LAZR system. Confocal images were captured using an Olympus FV3000 confocal microscope. FACS analysis was applied on a BD Calibur flow cytometer. High performance liquid chromatography (HPLC) experiments were made on a Waters Alliance e2695 system using a 2489 UV/Vis detector. Gel permeation chromatography (GPC) tests were recorded on a Thermo ULTIMATE 3000 instruments using a ERC RefratoMax520 refractive index detector.

## Synthesis and characterization

*RuDA*. A mixture of $[(\eta^6\text{-}p\text{-cym})Ru(phendio)Cl]Cl$ (phendio = 1,10- phenanthroline-5,6-dione)[64] (481.0 mg, 1.0 mmol), 4,7-bis[4-(N,N-diphenylamino)phenyl]-5,6-diamino-2,1,3-benzothiadiazole[65] (652.0 mg, 1.0 mmol) and glacial acetic acid (30 mL) was stirred at reflux temperature for 12 h. The solvent was then removed under vacuum with a rotary evaporator. The residue obtained was purified by flash column chromatography (silica gel, $CH_2Cl_2$: MeOH = 20:1) to give RuDA as green powder (yield: 877.5 mg, 80%). Anal. Calcd for $C_{64}H_{48}Cl_2N_8RuS$: C, 67.84; H, 4.27; N, 9.89. Found: C, 67.92; H, 4.26; N, 9.82. $^1H$ NMR (600 MHz, $d^6$-DMSO) δ 10.04 (s, 2H), 8.98 (s, 2H), 8.15 (s, 2H), 7.79 (s, 4H), 7.44 (s, 8H), 7.21 (d, $J$ = 31.2 Hz, 16H), 6.47 (s, 2H), 6.24 (s, 2H), 2.69 (s, 1H), 2.25 (s, 3H), 0.99 (s, 6H). $^{13}C$ NMR (150 MHz, $d^6$-DMSO), δ (ppm) 158.03, 152.81, 149.31, 147.98, 147.16, 139.98, 136.21, 135.57, 134.68, 130.34, 130.02, 128.68, 128.01, 125.51, 124.45, 120.81, 105.20, 103.49, 86.52, 84.75, 63.29, 30.90, 22.29, 18.83. ESI-MS: m/z $[M-Cl]^+$ = 1097.25.

*L2*. Synthesis of 4,7-bis[4-(N,N-diethylamino)phenyl-5,6-diamino-2,1,3-benzothiadiazole (**L2**): **L2** was synthesized in two steps. Pd(PPh$_3$)$_4$ (46 mg, 0.040 mmol) was added to N,N-diethyl-4-(tributylstannyl)aniline (1.05 g, 2.4 mmol) and 4,7-dibromo-5,6-dinitro-2,1,3-benzothiadiazole (0.38 g, 1.0 mmol) in dry toluene (100 mL). The mixture was stirred at 100 °C for 24 h. After removing the toluene under vacuum, the resulting solid was washed with petroleum ether. The mixture of this compound (234.0 mg, 0.45 mmol) and iron dust (0.30 g, 5.4 mmol) in acetic acid (20 mL) was then stirred at 80 °C for 4 h. The reaction mixture was poured into water and the resulting brown solid was collected by filtration. The product was purified by vacuum sublimation twice and obtained as a green solid (126.2 mg, 57% yield). Anal. Calcd for $C_{26}H_{32}N_6S$: C, 67.79; H, 7.00; N, 18.24. Found: C, 67.84; H, 6.95; N, 18.16. $.^1H$ NMR (600 MHz, CDCl$_3$), δ (ppm) 7.42 (d, 4H), 6.84 (d, 4H), 4.09 (s, 4H), 3.42 (d, 8H), 1.22(s, 12H). $^{13}C$ NMR (150 MHz, CDCl$_3$), δ (ppm) 151.77, 147.39, 138.07, 131.20, 121.09, 113.84, 111.90, 44.34, 12.77. ESI-MS: m/z $[M+H]^+$ = 461.24.

*RuET*. The compound was prepared and purified following a similar procedure as RuDA. Anal. Calcd for $C_{48}H_{48}Cl_2N_8RuS$: C, 61.27; H, 5.14; N, 11.91. Found: C, 61.32; H, 5.12; N, 11.81.$^1H$ NMR (600 MHz, $d^6$-DMSO), δ (ppm) 10.19 (s, 2H), 9.28 (s, 2H), 8.09 (s, 2H), 7.95 (s, 4H), 6.93(s, 4H), 6.48 (d, 2H), 6.34 (s, 2H), 3.54 (t, 8H), 2.80 (m, 1H), 2.33 (s, 3H), 1.31 (t, 12H), 1.07 (s, 6H). $^{13}C$ NMR (151 MHz, CDCl$_3$), δ (ppm) 158.20, 153.36, 148.82, 148.14, 138.59, 136.79, 135.75, 134.71, 130.44, 128.87, 128.35, 121.70, 111.84, 110.76, 105.07, 104.23, 87.10, 84.58, 44.49, 38.06, 31.22, 29.69, 22.29, 19.19, 14.98, 12.93. ESI-MS: m/z $[M-Cl]^+$ = 905.24.

*Photostability experiment*. RuDA was dissolved in MeOH/H$_2$O (5/95, v/v) at the concentration of 10 μM. Absorption spectra of RuDA were measured every 5 min on a Shimadzu UV-3600 spectrophotometer upon the irradiation of 808 nm laser (0.5 W cm$^{-2}$). The spectrum of ICG was recorded as the reference under the same condition.

*Electron spin resonance measurements*. The ESR spectra were recorded on a Bruker EMXmicro-6/1 spectrometer at 20 mW microwave power, 100 G-scan range, and 1 G field modulation. 2,2,6,6-Tetramethyl-4-piperidone (TEMP), and 5,5-dimethyl-1-pyrroline N-oxide (DMPO) were used as the spin-trapping agents. Mixed solutions of RuDA (50 μM) and TEMP (20 mM) or DMPO (20 mM) were under 808 nm laser irradiation (0.5 W cm$^{-2}$), and the electron spin resonance spectra were recorded.

*Computational details*. DFT and TD-DFT calculations of RuDA were carried out at the PBE1PBE/6–31 G*//LanL2DZ level in water solution by using the Gaussian 16 program[66–68]. The HOMO-LUMO and distribution of holes and electrons of the low-energy singlet excited states for RuDA were plotted using the GaussView program (version 5.0).

*Singlet oxygen ($^1O_2$) generation*. At first, we tried to determine the $^1O_2$ generation efficiency of RuDA by the traditional UV-vis spectrometry method with ICG ($\Phi_\Delta$ = 0.002) as a standard[51], however, the photodegradation of ICG greatly influences the results. Therefore, the $^1O_2$ quantum yield of RuDA was measured by detecting the fluorescence intensity change of ABDA at about 428 nm under 808 nm laser irradiation (0.5 W cm$^{-2}$). The experiment was carried out for RuDA and RuDA-NPs (20 μM) in water/DMF (98/2, v/v) containing ABDA (50 μM). The $^1O_2$ quantum yield was calculated according to the following equation: $\Phi_\Delta$ (PS) =

$\Phi_\Delta$ (ICG) × $(r_{PS}/A_{PS})/(r_{ICG}/A_{ICG})$. $r_{PS}$ and $r_{ICG}$ were the reaction rates of ABDA with $^1O_2$ obtained from photosensitizer and ICG, respectively. A$_{PS}$ and A$_{ICG}$ represent the absorbance of photosensitizer and ICG at 808 nm, respectively.

*Atomic force microscopy (AFM)*. AFM measurements were carried out on a Bruker Dimension Icon AFM system using scanasyst mode under liquid conditions. An open liquid cell configuration was used, and the cell was rinsed twice with ethanol, and dried with nitrogen flow. The dried cell was inserted into the optical head of the microscope. A drop of the sample was put in the liquid cell immediately, and on the cantilever using a sterile disposable plastic syringe and a sterile needle. Another drop was placed directly on the sample and when the optical head was lowered, the two droplets coalesced to form a meniscus between the sample and the liquid cell. AFM measurements were performed with V-shaped SCANASYST-FLUID nitride cantilevers (Bruker, spring constant k = 0.7 N m$^{-1}$, f$_0$ = 120–180 kHz).

*High performance liquid chromatography*. HPLC chromatograms were acquired on a Waters e2695 system equipped with a phoenix C18 column (250 × 4.6 mm, 5 μm) using a 2489 UV/Vis detector. The wavelength of the detector was 650 nm. Water and methanol were used as mobile phase A and B, respectively, and the flow rate of mobile phase was 1.0 mL min$^{-1}$. The gradient (solvent B) was as follows: 100% from 0 min to 4 min, 100% to 50% from 5 min to 30 min, reset to 100% from 31 min to 40 min. RuDA was dissolved in the mixed solution of methanol and water (50/50, v/v) at the concentration of 50 μM. The injection volume was 20 μL.

*Gel permeation chromatography*. GPC tests were recorded on a Thermo ULTIMATE 3000 instruments equipped with two PL aquagel-OH MIXED-H columns (2 × 300 × 7.5 mm, 8 μm) and an ERC RefratoMax520 refractive index detector. The GPC columns were eluted with water at 30 °C at the flow rate of 1 mL min$^{-1}$. RuDA-NPs were dissolved in PBS solution (pH = 7.4, 50 μM), and the injection volume was 20 μL.

*Measurement of photocurrent*. The photocurrent was determined on an electrochemical system (CHI-660B, China). The photoelectric response as laser light (808 nm, 0.5 W cm$^{-2}$) on and off were measured at 0.5 V in a black box, respectively. A standard three electrode cell with L-shaped glassy carbon electrode (GCE) as the working electrode, a standard calomel electrode (SCE) as the reference, and a platinum disk as the counter electrode were used. A solution of 0.1 M Na$_2$SO$_4$ was used as the electrolyte.

*Cell lines and cell culture*. Human breast cancer cell line MDA-MB-231 was purchased from KeyGEN Biotec Co., LTD (Nanjing, China, Cat. Number: KG033). The cells were grown monolayer in Dulbecco's modified Eagle medium (DMEM, high glucose) supplemented with 10% fetal bovine serum (FBS), penicillin (100 μg mL$^{-1}$), and streptomycin (100 μg mL$^{-1}$) solution. All cells were cultured in a humidified atmosphere containing 5% CO$_2$ at 37 °C.

*In vitro photocytotoxicity evaluation (MTT assays)*. MTT assays were used to determine the cytotoxicity of RuDA and RuDA-NPs in the presence and absence of light irradiation, with or without Vc (0.5 mM). MDA-MB-231 cancer cells were grown on 96-well plates at a cell density of $1 × 10^5$ cells/mL/well approximately, and incubated at an atmosphere of 5% CO$_2$ and 95% air at 37.0 °C for 12 h. RuDA and RuDA-NPs dissolved with water were added to the cells. After 12 h incubation, cells were exposed at 0.5 W cm$^{-2}$ for 10 min (300 J cm$^{-2}$) and incubated for 24 h in the dark. Then, cells were incubated an additional 5 h with MTT (5 mg mL$^{-1}$). Finally, the medium was replaced with DMSO (200 μL) to dissolve the resulting purple formazan crystals. The O.D. values were measured with a 570/630 nm enzyme-labeling instrument. The IC$_{50}$ value of each sample was calculated from the resulting dose-dependence curves by SPSS software from at least three independent experiments.

*Live/dead cell co-staining assay*. MDA-MB-231 cells were treated with RuDA and RuDA-NPs at a concentration of 50 μM. After 12 h incubation, the cells were irradiated with 808 nm laser at a power of 0.5 W cm$^{-2}$ for 10 min (300 J cm$^{-2}$). For the Vitamin C (Vc) group, the cells were treated with 0.5 mM Vc before laser irradiation. After that, the cells were incubated for a further 24 h in the dark, and then stained with calcein AM and propidium iodide (20 μg mL$^{-1}$, 5 μL) for 30 min, followed by washing with PBS (10 μL pH 7.4). Finally, the images of the stained cells were captured using a confocal microscope.

*Intracellular ROS generation*. MDA-MB-231 cells were treated with RuDA or RuDA-NPs at a concentration of 50 μM. After 12 h incubation, DCFH-DA dissolved in DMF (water-insoluble) was added to the cells for another 30 min incubation. Then, the cells were exposed to 808 nm laser at a power of 0.5 W cm$^{-2}$ for 10 min (300 J cm$^{-2}$). For the Vitamin C (Vc) group, the cells were treated with 0.5 mM Vc before laser irradiation. The fluorescence images of the cells were captured using a confocal microscope at 488 nm excitation.

*Cell apoptosis study by flow cytometry*. MDA-MB-231 cells were plated into 6-well culture plates (2 mL/well) and cultured in 5% CO$_2$ at 37 °C overnight. RuDA-NPs

or RuDA were added, which were diluted to a concentration of 50 μM. After 12 h incubation, the cells were irradiated with or without 808 nm laser (0.5 W cm$^{-2}$) for 10 min (300 J cm$^{-2}$). For the Vitamin C (Vc) group, the cells were treated with 0.5 mM Vc before laser irradiation. After another 12 h, the cells were digested with trypsin and washed twice with cold PBS. Then, cells were collected by centrifugation (450 × g, 5 min). The apoptosis was determined by flow cytometry using an Annexin V-FITC/PI assay kit (KeyGEN BioTECH, China) according to the manufacturer's protocol. The detailed operation as follows: cells were stained with 5 μL Annexin V-FITC for 5 min in Annexin-binding buffer (10 mM HEPES, 140 mM NaCl, 2.5 mM CaCl$_2$, pH 7.4). PI (propidium iodide) was added to cells with 5 μL before incubating at room temperature for 15 min. The fluorescence of cells was measured by flow cytometer. BD FACSDiva and Cell Quest Pro software were used for data collection and FlowJo V10 was used for data analysis. The results appeared as a percentage of normal and apoptotic cells at various stages.

*Mitochondrial membrane potential and superoxide generation assessment.* A total of 8 × 10$^5$ MDA-MB-231 cells were seeded in each well of 12-well plate for 24 h of incubation. Then, the cells were treated with RuDA-NPs or RuDA (50 μM) for 12 h and irradiated with or without 808 nm laser (0.5 W cm$^{-2}$) for 10 min (300 J cm$^{-2}$). After another 2 h of incubation, the cells were stained with JC-1 (5 μmol L$^{-1}$) for mitochondrial membrane potential and MitoSOX Red (5 μmol L$^{-1}$) for mitochondrial superoxide generation. Finally, the cells were visualized using Confocal laser scanning microscopy (CLSM) (Olympus FV3000, Japan) with 20×objective after washing with PBS twice.

*Subcellular distribution study.* MDA-MB-231 cancer cells were grown in 75 cm$^2$ flasks to approximately 50% confluency and treated with RuDA or RuDA-NPs at 50 μM for 12 h. Then, the cells were harvested and collected. Mitochondria were extracted from the cultured cells using a mitochondria isolation kit (Thermo Fisher Scientific) following the manufacturer's instructions. Nuclei of the MDA-MB-231 cells were isolated according to the previously established procedure[69]. The rest fractions were collected from the discarded fractions during the extraction of nuclei, and mitochondria. The contents of ruthenium were determined by using the inductively coupled plasma mass spectrometry (ICP-MS).

*Nrf2, HO-1, and Hsp70 expressions.* A total of 5 × 10$^6$ MDA-MB-231 cells were seeded into 6-well plates and incubated overnight. The cells were then treated for 12 h with RuDA-NPs (50 μM) or RuDA-NPs (50 μM) and irradiated with 808 nm laser (0.5 W, 10 min, 300 J cm$^{-2}$) using PBS as the control group. After that, the cells were lysed in cell lysis buffer (KeyGEN BioTECH, KGP701), and the whole proteins were obtained after centrifugation at 8050 × g for 5 min at 4 °C. A pre-stained protein molecular weight Marker, 20–120 kD, KeyGEN BioTECH, KGM441) was used to determine the different blots. Equal amounts of proteins were then transferred to nitrocellulose filter membranes and blocked with nonfat milk (5%) in TBST buffer at room temperature for 2 h. Then, the membranes were incubated with primary antibodies (anti-β-actin 1:1000, anti-NRF2 1:500, anti-HO-1 1:10000, and anti-HSP70 1:1000), washed for three times with cold TBST buffer, and finally incubated with secondary antibodies (Goat Anti-rabbit IgG, IgG-HRP, KeyGEN BioTECH, KGAA35, 1:500). The blots were visualized via a chemiluminescence light-based detector (G:BOX chemiXR5, SYNGENE) and analyzed with Gel-Pro 32 software. Full image is provided in the Source Data file. Antibody information: Rabbit anti-β-actin (ABCAM ab68226), anti-NRF2 (ABCAM ab62352), anti-HO-1 (ABCAM ab68477), and anti-HSP70 (ABCAM ab181606).

*Animals.* All animal experiments, surgical interventions, and postoperative animal care procedures were carried out under the guidelines approved by the Ethical Committee of Southeast University (Nanjing, China, permit no. SYXK20160014) and KeyGEN BioTECH Co. Ltd. (Nanjing, China, permit no. SYXK-20170040). *BALB/c* nude female mice were obtained from KeyGEN Biotec Co., LTD (Nanjing, China) and used for the construction of MDA-MB-231-bearing model. The animals were hosted in an equipped animal facility with temperature of 20–26 °C (daily temperature differences ≤ 4 °C) and humidity at 40–70%, under the same dark/light cycle (12 h:12 h). For the construction of MDA-MB-231 tumor-bearing mice, tumors grow to about 80 mm$^3$, which meets the maximum tumor size/burden permission by the ethics committees.

*PA imaging assessments.* MDA-MB-231 tumor (appr. 80 mm$^3$) bearing *BALB/c* nude female mice were anesthetized by isoflurane gas before PA imaging. The in vivo PA images and signal intensities at tumor sites were recorded from 0 to 24 h of post injection though tail veil with RuDA-NPs or RuDA (10 μmol kg$^{-1}$) with the irritation wavelength at 808 nm. The in vivo PA images of tumor sites after intratumoral injection of the nanoparticles of 5 μmol kg$^{-1}$ and 20 μmol kg$^{-1}$ were performed as control groups.

*Tissue biodistribution and excretion study.* MDA-MB-231 tumor bearing *BALB/c* nude female mice (tumor volume: appr. 80 mm$^3$) were injected with RuDA-NPs through the tail vein at a concentration of 10.0 μmol kg$^{-1}$ (n = 3 per group). After different time periods post-injection, the mice were sacrificed, and the major organs were collected to determine the ex vivo tissue biodistribution. Subsequently,

the extracted tissues were digested with 65% nitric acid and 30% H$_2$O$_2$ at 80 °C for 1 h. Finally, the samples were diluted to 2 mL with ultrapure water. After filtration, the concentrations of ruthenium were analyzed using ICP-MS.

For excretion study, the urine and feces samples of mice (n = 3) were collected in metabolic cages at different time points (0, 2, 4, 6, 8, and 10 days) after injection of RuDA or RuDA-NPs (10 μmol kg$^{-1}$) intravenously. Then, the samples were dissolved in 65% nitric acid and the amount of ruthenium was determined by ICP-MS.

*In vivo experiment.* For the construction of tumor-bearing mice, 100 μL of PBS with 3 × 10$^6$ MDA-MB-231 cells was injected into the axilla of *BALB/c* nude female mice. When the tumors grew to about 80 mm$^3$, the mice were randomly divided into different groups with 3 mice in each group. Animals were treated via intravenous injection through the tail vein with RuDA or RuDA-NPs (10.0 μmol kg$^{-1}$). The tumor site was irradiated under 808 nm laser (0.5 W cm$^{-2}$) for 10 min (300 J cm$^{-2}$) at 8 h post injection. Tumor volumes and body weights were measured every 3 days. Tumor volumes were obtained by measuring the perpendicular diameter of the tumor in length and width and calculated according to the formula: Tumor volume (mm$^3$) = 1/2 × length × width$^2$. At the last time point, the mice were sacrificed and the tumor tissues were excised and imaged.

*H&E staining.* For histological analysis, major organs (including heart, liver, spleen, lung, and kidney) and tumor tissues were harvested from killed mice on the 15th day post-treatment of different treatment conditions and fixed in 10% formalin. The samples were then dried and embedded in paraffin. Before immunostaining, tissue sections were dewaxed in xylene, rehydrated by gradient ethanol, washed with distilled water, and then stained with hematoxylin and eosin (H&E). After staining, the slices were dehydrated with increasing concentrations of ethanol and xylene. The morphological images of the tissues were captured using a light microscope.

*Statistical analysis.* All data were presented as mean ± s.d. unless stated otherwise. A two-sided Student's test was used to evaluate the statistical significance. *P* values < 0.05 represented statistically significant, *P* values: *$p < 0.05$, **$p < 0.01$, ***$p < 0.001$ (unpaired, two-sided t tests), and *p* values > 0.05 represented non-significance (N.S.). The exact *p* values are provided in the Source Data file.

**Reporting summary**. Further information on research design is available in the Nature Research Reporting Summary linked to this article.

## Data availability

The authors declare that all data supporting the findings of this study are available within the paper and its Supplementary Information files, including Supplementary Figures 1–33, and Supplementary Tables 1–5. The source data underlying Figs. 2B–D, 4B–D, 5C, D, 6A, B, F, 7A, C, 8B–D and Supplementary Figs. 6, 8–11, 13, 14, 23–26, 29, 31, 32 are provided with this paper as a Source Data file.

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

## Acknowledgements

We thank Prof. Wenjing Zhang for her helpful and valuable discussions on TD-DFT calculation. We thank the National Natural Science Foundation of China (Grant Nos. #22077015 and #21601034) and "Zhi-Shan" project of Southeast University for partly supporting this research. The authors are grateful to Jiangsu Province Hi-tech Key Laboratory for Biomedical Research for financial aid for this work.

## Author contributions

S.G., X.-H.X., and J.Z. oversaw and designed all experiments; G.X. and C.L. synthesized the complex; G.X., C.L., and Y. S. performed the biological evaluation; C.C. did the electrochemical experiments; G.X. and C.L. performed the animal experiments; L.W. did the PAI experiments; J.Z. and G.X. wrote the paper. All authors reviewed and edited the paper and have given approval to the final version of the manuscript.

## Competing interests

Southeast University has applied for a Chinese patent (Patent Number: CN 111808144 A) of RuDA reported here with J.Z. listed as one of the inventors. The remaining authors declare no competing interests.
