## [Peer Review File · Nature Communications]

REVIEWER COMMENTS

Reviewer #1 (Remarks to the Author): Expert in metal complexes, photodynamic therapy

Xia and co-workers present in their submission to Nature Communications. The authors must address the following issues carefully.

1. The author designed an arene-Ru(II) complex self-assembly photosensitizer for Near-infrared-activated phototherapy in this manuscript. In the previous published papers of authors (Adv. Funct. Mater. 2020, 2008325), self-assembles Iridium(III) complexes were also designed for NIR-triggered dual phototherapy. These two self-assembles complexes have almost identical design strategy, main ligand, optical properties, and so on, so what are the advantages and innovations in the present study?
2. The species of reactive oxygen species should be determined using ESR spectra.
3. "The 1O_2 quantum yield of RuDA is calculated to be 16.4% upon 808 nm irradiation", The solvent used should be added when describing quantum yield in the manuscript.
4. The long-term stability experiments of nanoparticles in PBS, FBS, and DMEM should be supplemented when nanoparticles are formed. The stability effect of different pH values on nanoparticles should also be added.
5. Only confocal fluorescence images for evaluating the ROS generation are displayed, the flow cytometer as a complementary technique should be added and used to quantitatively compare the amount of ROS produced in living cells.
6. The tissue distribution results indicated that 8 h post-injection is the optimum treatment based on only the inductively coupled plasma mass spectrometry method, Photoacoustic imaging or fluorescence imaging should also be added to further illustrate the process of RuDAD-NPs accumulation at the tumor site.
7. The arene-Ru(II) complex self-assembly photosensitizer as a nanoparticle for phototherapy in vivo, the therapeutic biosafety should be evaluated. the hematoxylin and eosin staining of the major organs should be added after treatment. The metabolism of RuDAD-NPs after post-injection through urine and faeces should also be discussed.
8. Hematoxylin and eosin staining of tumors with different treatments should be added to evaluate the therapeutic effect.
9. The concentration and irradiation parameters of photosensitizers and relevant dyes should be mentioned in the legends of the figures.
10. In Vivo experiment section, the approval number of animal use protocol should be indicated in this manuscript.
11. Only AFM image of RuDA in a CH₃OH/H₂O mixture with water fraction of 90% was done, the AFM image of RuDA in a CH₃OH should also be added as a control group. This quantitative comparison can more clearly support your conclusion.
12. The formatting of superscripts and subscripts should be carefully checked, such as "at ca. 1050 nm in CH₂Cl₂ and CH₃OH", "RuDA are determined to be 3.3% and 0.6% in CH₂Cl₂...", "(ABDA, a 1O_2 indicator)", "significant 1O_2 generation efficiency of RuDA in the aggregation state". Similar problems exist in this manuscript and should be checked carefully again.

Reviewer #2 (Remarks to the Author): Expert in sensitizers and cancer therapy

The manuscript entitled "A Highly Efficient Supramolecular Photosensitizer Derived from an Arene-Ru(II) Complex Self-assembly for Near-Infrared-Activated Photodynamic/Photothermal Therapy" submitted to Nature Communication by Zhao, Xia and Guo thematizes the development of Ru(II)-Arene complexes for near-infrared photodynamic and photothermal therapy. The presented topic is of high interest for the chemical community and is currently highly studied by various research groups around the world. Importantly, while the author note that there are no competing financial interest to declare, they have recently filed a patent on the here presented scientific content (Patent Number: CN 111808144; Application Number: CN 2020-10668118). The authors are required to comment on this and declare all potential conflicts of interests. In general, the study was proficiently carried out, although some important experiments are missing (see below for details). Unfortunately, the majority of experimental protocols are missing crucial information, making it impossible to reproduce the performed experiments. The authors should ensure to provide an in-depth description of all performed experiments. The English language and grammar could benefit from an in-depth proof reading. While the majority of the manuscript is understandable, some parts are poorly formulated, hampering their understanding. Overall, I recommend rejection of this manuscript due to the plurality of issues which are listed below:

- Ru(II) polypyridine complexes are receiving much attention as photosensitizers for photodynamic therapy. While some advantages of these compounds are mentioned, major studies in the field have not been mentioned or cited including but not limited to McFarland, Gasser, Chao, Glazer, Bonnet or Thomas. Importantly, the lead compound designed by McFarland is currently evaluated in phase II clinical trials as a novel photosensitizer. Such major developments in the field should be briefly mentioned in the introduction of this study.
- The authors have prepared a new Ru(II) complex with a significant absorption in the NIR region. As this is a highly desirable property, the authors should elaborate on their design of this metal complex so other scientists could learn from their design strategy to obtain the desired photophysical properties.
- The encapsulation of promising photosensitizers has recently received much attention. Unfortunately, the recent major studies in this field have not been mentioned or cited. Exemplary studies include but are not limited to recent studies by Gasser and Chao or Thomas. Highly relevant for the here described work, the previously mentioned authors have also reported on the encapsulation of Ru(II)polypyridine complexes with Pluronic F127, the same amphiphilic polymer as here described. These works should be described and contextualized to the here reported results.
- Some of the here described work has been presented in a separate section of the manuscript (Synthesis and Characterization) while the remaining part is presented as a continuous text. All results should be presented with the same formatting style.
- Experimental information on the synthesis of 4,7-bis[4-(N,N-diphenylamino)phenyl]-5,6-diamino-2,1,3-benzothiadiazole is missing.
- In the experimental section the synthesis of RuDA is based on "compound 2" as a metal precursor. Unfortunately, in the manuscript there is no reference to the structure of "compound 2". The authors should ensure that the experimental protocol is understandable for the reader and contains all necessary information for the reproduction of the here presented experiments.
- The aromatic system of p-cymene in Scheme 1 is missing a double bond to form the aromatic system.

- As a potential photophysical mechanism, the HOMO and LUMO orbitals of the metal complexes were calculated. While both MOs are centered on the extended organic ligand this does not provide enough evidence for the by the author suggested hypothesis of an intermolecular charge transfer transition. For such investigations, the authors would need to study all the accessible excited state conformation and transition of the monomer and oligomeric aggregate in-depth by TD-DFT calculations.
- The size of the generated nanoparticles was characterized using DLS. While the plot shows a uniform distribution, more data points especially towards bigger particle sizes are necessary for a better understanding of the particle distribution. In addition, the polydispersity of the mixture needs to be determined.
- The photostability of RuDA was studied over a time frame of 0-25 min. In the experimental section it is noted that the whole absorption spectrum is monitored. Unfortunately, the spectrum is not shown and only a normalized value at one wavelength. The whole spectrum should be recorded and shown (maybe in the supporting information) to ensure that none of the other parts are changing. In addition, beside monitoring in dependence of the irradiation time also the light dose needs to be studied as a crucial factor. For a meaningful insight, the authors would need to investigate which light dose the here studied time frame corresponds to. They should ensure to reach at least the clinically applied levels (within the range of 200 J/cm²).
- In general, the authors should mention in every biological experiment, the applied light dose as a crucial parameter.
- Ruthenium-Arene complexes are well-known to degrade within physiological conditions (see works of Dyson, Hartinger or Keppler). The stability of the reported compounds needs to be investigated in-depth to ensure understanding which components cause which biological effects.
- The authors refer to an inhibition effect upon irradiation of the photosensitizer. However, no specific cell mechanism is inhibited and instead cytotoxic species generated which cause cell death. This terminology needs to be corrected.
- The biodistribution of the metal complex is missing as a reference to the injected dose. The presented data in Figure 5A should be normalized towards the injected dose.
- As an important factor, the subcellular localization as well cell death mechanism of the photosensitizer has a drastic effect on the photodynamic efficacy. For a full biological evaluation of the here reported nanoparticles, the sub-organelle localization and cell death mechanism needs to be investigated.

Reviewer #1 (Remarks to the Author): Expert in metal complexes, photodynamic therapy:

Xia and co-workers present in their submission to *Nature Communications*. The authors must address the following issues carefully.

1. The author designed an arene-Ru(II) complex self-assembly photosensitizer for Near-infrared-activated phototherapy in this manuscript. In the previous published papers of authors (*Adv. Funct. Mater.* **2020**, 2008325), self-assembles Iridium(III) complexes were also designed for NIR-triggered dual phototherapy. These two self-assembles complexes have almost identical design strategy, main ligand, optical properties, and so on, so what are the advantages and innovations in the present study?

Answer: We thank this reviewer for the constructive comment to highlight the novelty of our submitted manuscript.

Different from the conventional metal-based photosensitizers, this series of metal-based photosensitizers exhibit unique photophysical and photochemical properties, including broad absorption in the NIR region, aggregation-induced $^1\text{O}_2$ generation, and synergistic PDT and PTT effects, which are highly desired for phototherapy. Although a NIR-triggered supramolecular iridium(III) complex has been designed before, the underlying mechanisms of the properties of the complex remain unclear. Moreover, to the best of our knowledge, it is the first example of an organometallic Ru(II)-arene complex could be excited with one photon excitation at 808 nm for dual phototherapy, which extends the application of Ru(II)-arene complexes not only in chemotherapy but also in phototherapy. Herein, two novel arene-Ru(II) complexes including a contrast compound were deliberately designed for comparison, and clearly elucidated the underlying mechanisms on the basis of DFT calculation and electrochemical investigations. The present study can provide useful insight into the molecular design of NIR-excited organometallic Ru(II)-arene photosensitizers based on their spatial structure and ionic character for potential clinical applications.

2. The species of reactive oxygen species should be determined using ESR spectra.

Answer: We thank this reviewer for your professional advice. In this revised manuscript, electron spin resonance (ESR) spectroscopy with 2,2,6,6-tetramethyl-4-piperidone (TEMP) and 5,5-dimethyl-1-pyrroline N-oxide (DMPO) as the spin-trapping agents was utilized to identify the ROS species generated by RuDA. As shown in **Figure R1**, the enhanced triplet ESR signal of 2,2,6,6-tetramethyl-4-piperidone-N-oxyl (TEMPO) generated by the reaction of TEMP with $^1\text{O}_2$ was observed as the irradiation time increased from 0 to 4 min, confirming the generation of $^1\text{O}_2$. Besides, the typical 1:2:2:1 four-line ESR signal of DMPO-OH \cdot adducts was detected when RuDA was incubated with DMPO upon irradiation, demonstrating the generation of hydroxyl radical (OH \cdot). These results indicate the ability of RuDA to promote the ROS generation *via* a dual type I/II photosensitization process.

Figure R1. EPR signals of TEMPO and DMPO- $\text{OH}\cdot$ adducts for (A) $^1\text{O}_2$ and (B) $\cdot\text{OH}$ characterization upon NIR (808 nm, 0.5 W cm^{-2}) irradiation of the mixture of RuDA (50 μM) and TEMP (20 mM) or DMPO (20 mM) at 0, 2, and 4 min, respectively.

3. “The $^1\text{O}_2$ quantum yield of RuDA is calculated to be 16.4% upon 808 nm irradiation”, The solvent used should be added when describing quantum yield in the manuscript.

Answer: We are sorry for our negligence. The $^1\text{O}_2$ quantum yield of RuDA was investigated in 98% $\text{H}_2\text{O}/2\%$ DMF due to the water-insoluble of ABDA. The related information has been added to the text.

4. The long-term stability experiments of nanoparticles in PBS, FBS, and DMEM should be supplemented when nanoparticles are formed. The stability effect of different pH values on nanoparticles should also be added.

Answer: As suggested by this reviewer, the stability of RuDA-NPs was tested in PBS (pH = 5.4, 7.4 and 9.0), FBS, and DMEM using UV-vis absorption spectra at different time points. As exhibited in **Figure R2**, negligible changes of the absorption bands of RuDA-NPs in PBS (pH = 7.4 and 9.0), FBS, and DMEM were observed, indicating the good stability of RuDA-NPs. However, the hydrolysis of RuDA was detected under acidic condition (pH = 5.4, **Figure R2A**).

Figure R2. UV-vis absorption spectra of RuDA-NPs in PBS (A) pH=5.4, (B) pH=7.4, (C) pH=9.0, (D) FBS, and (E) DMEM at various time points.

5. Only confocal fluorescence images for evaluating the ROS generation are displayed, the flow cytometer as a complementary technique should be added and used to quantitatively compare the amount of ROS produced in living cells.

Answer: As suggested by this reviewer, we further determined the intracellular ROS levels in RuDA-NPs- and RuDA-treated MDA-MB-231 cells by flow cytometry (**Figure R3A**). As shown in **Figure R3B**, RuDA-NPs and RuDA under 808 nm irradiation produced a significant increase of the mean fluorescence intensity (MFI) by approximate 5.1- and 4.8-fold than control group, respectively, confirming the excellent ROS generation abilities of RuDA-NPs and RuDA. However, the intracellular ROS levels in RuDA-NPs- and RuDA-treated MDA-MB-231 cells are comparable to the control group in the absence of laser or in the presence of Vc, which is similar to the result of confocal fluorescence analysis.

Figure R3. (A) Flow cytometry analysis of ROS levels in MDA-MB-231 cells under different treatment conditions; (B) Quantitative analysis of the mean fluorescent intensity (MFI) of DCF. The results are mean \pm SD (n = 3). (**) $p < 0.01$ compared with the value of the control group (PBS + irradiation).

6. The tissue distribution results indicated that 8 h post-injection is the optimum treatment based on only the inductively coupled plasma mass spectrometry method, Photoacoustic imaging or fluorescence imaging should also be added to further illustrate the process of RuDA-NPs accumulation at the tumor site.

Answer: Thanks for the constructive suggestion. We carried out the photoacoustic (PA) performance of RuDA-NPs by recording the PA signals of RuDA-NPs at different post-injection times. First, the *in vivo* PA signals of RuDA-NPs were evaluated by recording the PA images of tumor sites after intratumoral injection of RuDA-NPs. As shown in **Figure R4**, RuDA-NPs displayed strong PA signals, and there is a positive correlation between the concentration of the RuDA-NPs and the intensity of the PA signals. Then, the *in vivo* PA images of tumor sites after intravenous injection of RuDA and RuDA-NPs were recorded at varied time points of post injection. As shown in **Figure R5**, the PA signals of RuDA-NPs from the tumor sites gradually enhanced with time, and reached a plateau at 8 h of post injection, which is in accordance with the tissue distribution results determined by ICP-MS analysis. As for RuDA (**Figure R6**), the maximum PA signal intensity appeared at 4 h of post injection, implying a rapid tumor penetration rate of RuDA.

Figure R4. *In vivo* PA images of tumor sites under excitation at 808 nm after intratumoral injection of RuDA-NPs at different concentrations ($5 \mu\text{mol kg}^{-1}$, $20 \mu\text{mol kg}^{-1}$).

Figure R5. *In vivo* PA images of tumor sites under excitation at 808 nm after intravenous injection of RuDA-NPs ($10 \mu\text{mol kg}^{-1}$) at different time points.

Figure R6. *In vivo* PA images of tumor sites under excitation at 808 nm after intravenous injection of RuDA ($10 \mu\text{mol kg}^{-1}$) at different time points.

7. The arene-Ru(II) complex self-assembly photosensitizer as a nanoparticle for phototherapy *in vivo*, the therapeutic biosafety should be evaluated. The hematoxylin and eosin staining of the major organs should be added after treatment. The metabolism of RuDA-NPs after post-injection through urine and faeces should also be discussed.

Answer: We thanks this reviewer for the valuable advice. Based on your suggestion, the hematoxylin and eosin staining of the major organs (heart, liver, spleen, lung, and kidney) from different treatment groups were recorded. As shown in **Figure R7**, the H&E staining images of five major organs from RuDA-NPs and RuDA groups did not exhibit obvious abnormalities or organ damages. These results demonstrate that both RuDA-NPs and RuDA had no sign of *in vivo* toxicity.

Figure R7. H&E staining images of major organs and tumors from different treatment groups, including Saline, Saline + Laser, RuDA, RuDA + Laser, RuDA-NPs, and RuDA-NPs + Laser groups. Scale bars: 60 μm .

The excretion behaviors of RuDA and RuDA-NPs were also investigated by determining the ruthenium amount in urine and feces using ICP-MS. As shown in **Figure R8**, the primary excretion pathway of RuDA and RuDA-NPs was through feces, and efficient clearance of RuDA and RuDA-NPs was observed within the study period of 8 d, implying that RuDA and RuDA-NPs could be effectively excreted from the body without long-term toxicity.

Figure R8. Ru excreted out of the mice body *via* urine and feces after intravenous administration of (A) RuDA ($10 \mu\text{mol kg}^{-1}$) and (B) RuDA-NPs ($10 \mu\text{mol kg}^{-1}$) at different time intervals. The data represent the mean \pm SD ($n=3$).

8. Hematoxylin and eosin staining of tumors with different treatments should be added to evaluate the therapeutic effect.

Answer: According to the comment, the H&E staining images of tumors from different treatment groups were recorded to evaluate the therapeutic efficacy. As shown in the right lane of **Figure R7**,

both RuDA + Laser and RuDA-NPs + Laser groups could cause severe cancer cell destruction, confirming the excellent photo-induced anticancer activity of RuDA and RuDA-NPs.

9. The concentration and irradiation parameters of photosensitizers and relevant dyes should be mentioned in the legends of the figures.

Answer: We are sorry for our negligence. The concentration and irradiation parameters have been added to the legends of the figures.

10. *In Vivo* experiment section, the approval number of animal use protocol should be indicated in this manuscript.

Answer: Thank you very much for your kind reminder. The approval number of animal use protocol was added in the manuscript.

The animal procedure followed the Institutional Animal Care and Use Committee at KeyGEN BioTECH Co. Ltd. (Nanjing, China, approval no. SYXK-2017-0040) and Ethical Committee of Southeast University (Nanjing, China, approval no. SYXK-2016-0014).

11. Only AFM image of RuDA in a CH₃OH/H₂O mixture with water fraction of 90% was done, the AFM image of RuDA in a CH₃OH should also be added as a control group. This quantitative comparison can more clearly support your conclusion.

Answer: We thank this reviewer for this helpful suggestion. The in-liquid AFM image of RuDA in methanol was added to the revised manuscript. As shown in **Figure R9**, no obvious aggregation was observed.

Figure R9. In-liquid AFM image of RuDA in CH₃OH.

12. The formatting of superscripts and subscripts should be carefully checked, such as “at ca. 1050 nm in CH₂Cl₂ and CH₃OH”, “RuDA are determined to be 3.3% and 0.6% in CH₂Cl₂...”, “(ABDA, a ¹O₂ indicator)”, “significant ¹O₂ generation efficiency of RuDA in the aggregation state”. Similar problems exist in this manuscript and should be checked carefully again.

Answer: Thanks for your careful review. We are sorry for the negligence. We have carefully checked the whole manuscript and corrected the errors (marked in red).

Reviewer #2 (Remarks to the Author): Expert in sensitizers and cancer therapy

The manuscript entitled "A Highly Efficient Supramolecular Photosensitizer Derived from an Arene-Ru(II) Complex Self-assembly for Near-Infrared-Activated Photodynamic/Photothermal Therapy" submitted to Nature Communication by Zhao, Xia and Guo thematizes the development of Ru(II)-Arene complexes for near-infrared photodynamic and photothermal therapy. The presented topic is of high interest for the chemical community and is currently highly studied by various research groups around the world. Importantly, while the author note that there are no competing financial interest to declare, they have recently filed a patent on the here presented scientific content (Patent Number: *CN 111808144*; Application Number: *CN 2020-10668118*). The authors are required to comment on this and declare all potential conflicts of interests. In general, the study was proficiently carried out, although some important experiments are missing (see below for details). Unfortunately, the majority of experimental protocols are missing crucial information, making it impossible to reproduce the performed experiments. The authors should ensure to provide an in-depth description of all performed experiments. The English language and grammar could benefit from an in-depth proof reading. While the majority of the manuscript is understandable, some parts are poorly formulated, hampering their understanding. Overall, I recommend rejection of this manuscript due to the plurality of issues which are listed below:

Answer: We appreciate the reviewer's time and effort on reviewing our manuscript. We hope our explanation below can address your concerns.

*Thank you for your kind reminder. It is true that one of the corresponding author Jian Zhao has applied for a Chinese patent (Patent Number: *CN 111808144A*) on some of the compounds reported here. The compounds were synthesized by Jian Zhao, and the related biological evaluation, DFT calculation and electrochemical investigation performed in this manuscript were carried out by the collaboration of Xia and Gou groups, who declare no conflict of interest. Besides, under Chinese patent law, the patent is a service invention and belongs to the employer (Southeast University). Furthermore, no author in this manuscript works for any pharmaceutical companies or R&D centers. The related declaration has been added to the manuscript.*

Besides, we have made the experimental section complete, and all the experiments we performed are repeatable.

- Ru(II) polypyridine complexes are receiving much attention as photosensitizers for photodynamic therapy. While some advantages of these compounds are mentioned, major studies in the field have not been mentioned or cited including but not limited to McFarland, Gasser, Chao, Glazer, Bonnet or Thomas. Importantly, the lead compound designed by McFarland is currently evaluated in phase II clinical trials as a novel photosensitizer. Such major developments in the field should be briefly mentioned in the introduction of this study.

Answer: Thank you for your valuable suggestions. Upon great efforts of many research groups including those you mentioned, great progresses have been made in the development of Ru(II)-based photosensitizers for clinical application. The notable example is TLD-1433 that was developed by McFarland, which is the first Ru(II)-based photosensitizer to enter a human clinical trial. These progresses are very encouraging and important. According to your suggestions, we have revised the introduction and the related references (28-41) have been supplemented.

- The authors have prepared a new Ru(II) complex with a significant absorption in the NIR region. As this is a highly desirable property, the authors should elaborate on their design of this metal complex so other scientists could learn from their design strategy to obtain the desired photophysical properties.

Answer: Thank you for your professional comment and suggestion. The photophysical properties of the reported Ru(II) complex, including NIR absorption, aggregation-induced PDT behavior and efficient photothermal performance, are attractive features for application as a photosensitizer. It is well known that the donor-acceptor (D-A) type dye usually has a narrow band gap that can result in a powerful absorption capability of long-wavelength light. However, these dyes are generally insoluble or weakly soluble in common solvents, such as tetrahydrofuran (THF), dichloromethane (DCM), and chloroform, even in DMF and DMSO. Thus, we speculate that the ionic character of the Ru(II)-based complexes can improve the solubility of the D-A chromophores in common solvents and enhance the assembly of the D-A chromophores. Moreover, the pseudo-octahedral half-sandwich structure of the organometallic Ru(II)-arene complex could sterically prevent H-aggregation of the chromophores, thus favoring the formation of J-aggregation with a red-shifted absorption band.

As for the aggregation-induced PDT behavior, this result has exceeded our expectation. Thus, the underlying mechanism were explored by DFT calculations and electrochemical analysis. The aggregation of D-A compounds we think can facilitate the ISC process *via* the dye molecular electronic coupling and interaction, which has the potential to increase the possible ISC transition channels and minimize the energy gap (ΔE_{ST}) between the singlet and triplet excited states, thus promoting the ISC efficiency to facilitate 1O_2 production. The related information has been added to the revised manuscript.

- The encapsulation of promising photosensitizers has recently received much attention. Unfortunately, the recent major studies in this field have not been mentioned or cited. Exemplary studies include but are not limited to recent studies by Gasser and Chao or Thomas. Highly relevant for the here described work, the previously mentioned authors have also reported on the encapsulation of Ru(II) polypyridine complexes with Pluronic F127, the same amphiphilic polymer as here described. These works should be described and contextualized to the here reported results.

Answer: Some nice work has been done by Thomas, Gasser and Chao in this field. The related articles are of great significance and helpful to highlight the importance of this this research filed. The background information and references have been mentioned in the revised manuscript and the related references (46-49) has been added to the manuscript.

- Some of the here described work has been presented in a separate section of the manuscript (Synthesis and Characterization) while the remaining part is presented as a continuous text. All results should be presented with the same formatting style.

Answer: Thank you for your kind reminder. The section of "Synthesis and Characterization" has been combined with the "Results and Discussion".

- Experimental information on the synthesis of 4,7-bis[4-(N,N-diphenylamino)phenyl]-5,6-diamino-2,1,3-benzothiadiazole is missing.

Answer: Sorry for our negligence. 4,7-Bis[4-(N,N-diphenylamino)phenyl]-5,6-diamino-2,1,3-

benzothiadiazole was synthesized by a literature method (*Adv. Mater.* **2009**, 21, 111-116), which has been added to the references (Ref 65).

- In the experimental section the synthesis of RuDA is based on “compound 2” as a metal precursor. Unfortunately, in the manuscript there is no reference to the structure of “compound 2”. The authors should ensure that the experimental protocol is understandable for the reader and contains all necessary information for the reproduction of the here presented experiments.

Answer: Compound 2 is $[(\eta^6\text{-}p\text{-cymene})\text{Ru}(\text{phendio})\text{Cl}]\text{Cl}$ (phendio = 1,10-phenanthroline-5,6-dione), whose structure was shown in **Scheme S1**. Compound 2 was replaced with $[(\eta^6\text{-}p\text{-cym})\text{Ru}(\text{phendio})\text{Cl}]\text{Cl}$ (phendio = 1,10-phenanthroline-5,6-dione) in the revised manuscript.

- The aromatic system of p-cymene in Scheme 1 is missing a double bond to form the aromatic system.

Answer: For the preparation of the dimer $[\text{Ru}(\eta^6\text{-}p\text{-cymene})\text{Cl}_2]_2$, α -terpinene or α -phellandrene instead of p-cymene was used as starting materials to react with RuCl_3 .

- As a potential photophysical mechanism, the HOMO and LUMO orbitals of the metal complexes were calculated. While both MOs are centered on the extended organic ligand this does not provide enough evidence for the by the author suggested hypothesis of an intermolecular charge transfer transition. For such investigations, the authors would need to study all the accessible excited state conformation and transition of the monomer and oligomeric aggregate in-depth by TD-DFT calculations.

Answer: Thank you for your valuable suggestions. By using TD-DFT calculations, the distribution of electrons and holes for the low-energy singlet excited states of RuDA in both monomeric and dimeric forms was analyzed to get deep insight into the electronic characters of the excited states (**Tables R1 and R2**). Notably, a proportion of intermolecular CT character was observed in most of these singlet states, especially for S_3 and S_4 , which are dominated by intermolecular CT transition based on intermolecular CT analysis (**Table R3**). To better understand the experimental results, we further probe the excited states of RuDA in the monomer and the dimer to explicit their difference. The transition configurations of the singlet and triplet excited states in monomeric and dimeric RuDA were revealed by TD-DFT calculations (**Tables R4-R5**). As shown in **Figure R10**, only one ISC channel is observed in the monomer. However, four ISC channels are present in the dimeric form, which could enhance the ISC transition. Therefore, it is reasonable to infer that the more RuDA molecules aggregate, the more ISC channels are accessible. Consequently, the aggregates of RuDA can form two band-like electronic structures in both singlet and triplet states with decreased energy gap between S_1 and available T_n , promoting the ISC efficiency to facilitate $^1\text{O}_2$ production.

Table R1. Distribution of holes, electrons, and overlaps of the low-energy singlet excited states for RuDA in the monomeric form.

Excited state and properties	Holes	Electrons	Overlap
S ₁ 1.30 eV $f = 0.4228$			S ₂ 1.64 eV $f = 0.0212$			S ₃ 2.28 eV $f = 0.0012$			S ₄ 2.39 eV $f = 0.0001$			S ₅ 2.61 eV $f = 0.0128$			S ₆ 2.67 eV $f = 0.0449$			
Table R2. Distribution of holes, electrons, and overlaps of the low-energy singlet excited states for RuDA in the dimeric form.

Excited state and properties	Holes	Electrons	Overlap
S ₁ 1.27 eV $f = 0.0280$			S ₂ 1.28 eV $f = 0.6370$			S ₃ 1.29 eV $f = 0.0232$			S ₄ 1.31 eV $f = 0.0299$			S ₅ 1.59 eV $f = 0.0316$			S ₆ 1.61 eV $f = 0.0040$			S ₇ 1.69 eV $f = 0.0000$			

Table R3. Contribution ratio (η) of RuDA in the dimeric form for the electron transition from one RuDA molecule to the other one in the low-energy singlet excited states.

	S ₁	S ₂	S ₃	S ₄	S ₅	S ₆	S ₇	S ₈	S ₉	S ₁₀
η (%)	2.5	0.6	95.6	93.4	0.1	0	99.8	99.7	0.2	0.3

Table R4. The transition configurations of the singlet and triplet excited states in the monomeric RuDA calculated by TD-DFT calculations.

	n	Energy (eV)	Transition configuration
S _n	1	1.2987	H → L (100%)
	2	1.6352	H-1 → L (99%)
	3	2.2761	H → L+1 (99%)
	4	2.3938	H-2 → L (98%)
	5	2.6052	H-1 → L+1 (61%), H → L+2 (38%)
	6	2.6655	H-1 → L+1 (39%), H → L+2 (59%)
	7	2.7410	H-3 → L (86%), H-3 → L+1 (7%), H-3 → L+4 (5%)
	8	2.7608	H → L+3 (80%), H → L+4 (15%)
	9	2.7813	H-17 → L (18%), H-9 → L (23%), H-5 → L (54%)
	10	2.8287	H-3 → L (11%), H-3 → L+1 (13%), H-3 → L+3 (18%), H-3 → L+4 (41%), H-4 → L+5 (7%)
	11	2.9302	H-4 → L (29%), H-4 → L+1 (13%), H-4 → L+4 (35%), H-4 → L+3 (9%)
	12	2.9821	H-4 → L (66%), H-4 → L+3 (10%), H-4 → L+4 (15%)
	13	2.9940	H-1 → L+2 (97%)
	14	3.0683	H → L+3 (15%), H → L+4 (83%)

	15	3.0861	H-11 → L (11%), H-7 → L (83%), H-10 → L (3%)
	16	3.1641	H-1 → L+3 (82%), H-1 → L+4 (12%)
	17	3.1835	H-3 → L+5 (66%) H-20 → L+5 (3%), H-12 → L+4 (5%), H-4 → L+1 (2%), H-3 → L+2 (7%), H-3 → L+8 (2%)
	18	3.2308	H-17 → L (40%), H-5 → L (38%) H-16 → L (5%), H-15 → L (2%), H-9 → L (8%), H-8 → L (2%)
	19	3.2446	H-10 → L (12%), H-6 → L (81%), H-11 → L (2%)
	20	3.2845	H-3 → L+1 (74%), H-3 → L+4 (13%), H-4 → L+5 (4%), H-2 → L+1 (3%)
T _n	1	0.6494	H-2 → L (24%), H → L (75%)
	2	1.4309	H-1 → L (98%)
	3	1.7199	H-2 → L (65%), H → L (26%), H-16 → L (3%)
	4	2.2025	H-3 → L+1 (10%), H-3 → L+3 (19%), H-3 → L+4 (57%), H-20 → L+4 (3%), H-4 → L+5 (2%)
	5	2.2430	H → L+1 (91%), H → L+3 (3%)
	6	2.3714	H-17 → L (21%), H-9 → L (15%), H-5 → L (21%), H → L+3 (16%), H-2 → L+3 (5%), H → L+1 (4%), H → L+4 (4%)
	7	2.4130	H-4 → L+3 (18%), H-4 → L+4 (51%), H-19 → L+4 (2%), H-4 → L+1 (9%), H-4 → L+7 (2%), H-3 → L+5 (4%)
	8	2.4669	H → L+2 (67%), H-15 → L (2%), H-4 → L (3%), H-2 → L+2 (5%), H-1 → L+1 (4%), H → L+3 (3%)
	9	2.4728	H-5 → L (11%), H → L+3 (29%), H-18 → L (3%), H-17 → L (7%), H-16 → L (3%), H-9 → L (7%), H-2 → L+3 (7%), H → L+2 (6%), H → L+4 (8%)
	10	2.5575	H-3 → L+5 (73%), H-20 → L+5 (5%), H-4 → L+4 (3%), H-3 → L+2 (5%), H-3 → L+8 (3%)
	11	2.6158	H-1 → L+1 (87%) H-15 → L (5%), H-4 → L (3%)
	12	2.6861	H-15 → L (35%), H-4 → L (26%), H → L+2 (12%)
	13	2.7264	H-3 → L (83%), H-3 → L+1 (5%)
	14	2.8013	H-4 → L+5 (62%), H-15 → L+5 (4%), H-12 → L+5 (3%), H-4 → L+2 (6%), H-4 → L+8 (3%), H-3 → L (4%)
	15	2.8416	H-1 → L+2 (20%), H → L+6 (23%), H-18 → L (4%), H-18 → L+1 (2%), H-4 → L+5 (5%), H-3 → L (5%), H-1 → L+9 (3%), H → L+3 (4%)
	16	2.8694	H-12 → L+4 (23%), H-12 → L+5 (11%), H-19 → L+3 (2%), H-19 → L+4 (7%), H-19 → L+5 (7%), H-15 → L+3 (2%), H-15 → L+4 (7%), H-15 → L+5 (2%), H-12 → L+1 (3%), H-12 → L+3 (8%), H-12 → L+7 (2%), H-4 → L+5 (4%)
	17	2.9216	H-16 → L (10%), H → L+6 (25%), H-18 → L (3%), H-18 → L+1 (7%), H-16 → L+1 (2%), H-3 → LUMO (5%), H-3 → L+1 (7%), H → L+3 (4%)
	18	2.9496	H-19 → L+5 (13%), H-12 → L+4 (10%), H-12 → L+5 (30%) H-19 → L+4 (3%), H-15 → L+4 (3%), H-15 → L+5 (8%), H-12 → L+1

			(2%), H-12 → L+3 (3%), H-12 → L+8 (2%)
	19	2.9705	H-4 → L (12%), H-1 → L+3 (26%), H-1 → L+6 (10%), H-11 → L (3%), H-7 → L (9%), H-1 → L+4 (6%), H → L+9 (7%), H → L+15 (5%)
	20	2.9915	H-11 → L (11%), H-7 → L (60%), H-10 → L (5%), H-4 → L (9%), H-1 → L+2 (3%)

Table R5. The transition configurations of the singlet and triplet excited states in the dimeric RuDA calculated by TD-DFT calculations.

	n	Energy (eV)	Transition configuration
S _n	1	1.2702	H-1 → L (71%), H → L (3%), H → L+1 (24%)
	2	1.2829	H-1 → L (24%), H → L (22%), H → L+1 (51%)
	3	1.2939	H-2 → L (12%), H → L (62%), H → L+1 (23%)
	4	1.3053	H-3 → L+1 (14%), H-1 → L+1 (83%), H-3 → L (2%)
	5	1.5898	H-3 → L (32%), H-2 → L (10%), H-2 → L+1 (53%), H-3 → L+1 (5%)
	6	1.6071	H-3 → L (58%), H-2 → L+1 (34%), H-3 → L+1 (4%), H-2 → L (3%)
	7	1.6892	H-2 → L (73%), H-2 → L+1 (11%), H → L (12%)
	8	1.7034	H-3 → L+1 (75%), H-1 → L+1 (16%), H-3 → L (7%)
	9	2.2714	H → L+3 (99%)
	10	2.2784	H-1 → L+2 (98%)
	11	2.3360	H-3 → L+3 (12%), H-1 → L+3 (88%)
	12	2.3567	H-5 → L (31%), H-4 → L (50%), H-4 → L+1 (16%)
	13	2.3693	H-5 → L+1 (61%), H-4 → L (17%), H-4 → L+1 (18%)
	14	2.4161	H → L+2 (96%) H-2 → L+2 (3%)
	15	2.5120	H-5 → L+1 (27%), H-4 → L+1 (62%), H-5 → L(4%), H-4 → L (6%)
	16	2.5175	H-5 → L (61%), H-5 → L+1 (10%), H-4 → L (25%), H-4 → L+1 (3%)
	17	2.5814	H-2 → L+3 (73%), H → L+5 (24%)
	18	2.5935	H-3 → L+2 (69%), H-1 → L+4 (29%)
	19	2.6492	H-2 → L+3 (25%), H → L+5 (67%), H → L+4 (3%)
	20	2.6563	H-3 → L+2 (29%), H-1 → L+4 (64%)
T _n	1	0.6479	H-4 → L (18%), H-1 → L (66%), H-5 → L (9%), H-4 → L+1 (3%), H-1 → L+1 (8%)
	2	0.6480	H-5 → L+1 (19%), H → L+1 (67%), H-4 → L+1 (7%), H → L (8%)
	3	1.2930	H-2 → L (13%), H → L (76%), H → L+1 (9%)
	4	1.3022	H-3 → L+1 (16%), H-1 → L+1 (73%), H-1 → L (9%)
	5	1.3992	H-2 → L (11%), H-2 → L+1 (87%)
	6	1.4267	H-3 → L (87%), H-3 → L+1 (10%)
	7	1.6798	H-5 → L (11%), H-4 → L (35%), H-2 → L (12%), H-1 → L (21%), H-4 → L+1 (4%), H → L (3%)
	8	1.6830	H-5 → L+1 (40%), H-4 → L+1 (16%), H → L+1 (24%), H-5 → L (5%), H → L (3%)

9	1.6909	H-2 → L (62%), H-5 → L (5%), H-4 → L (5%), H-2 → L+1 (8%), H-1 → L (5%), H → L (9%)
10	1.7031	H-3 → L+1 (71%), H-1 → L+1 (16%), H-3 → L (8%)
11	2.1915	H-7 → L+2 (10%), H-7 → L+6 (15%), H-7 → L+8 (52%), H-41 → L+8 (2%), H-7 → L+9 (9%)
12	2.1971	H-6 → L+3 (10%), H-6 → L+7 (15%), H-6 → L+9 (52%), H-40 → L+9 (2%), H-6 → L+8 (9%)
13	2.2409	H → L+3 (92%), H → L+7 (2%)
14	2.2484	H-1 → L+2 (92%), H-1 → L+6 (2%)
15	2.3356	H-3 → L+3 (12%), H-1 → L+3 (88%)
16	2.3669	H-34 → L+1 (20%), H-19 → L+1 (10%), H-10 → L+1 (13%), H → L+7 (15%), H-34 → L (2%), H-18 → L+1 (3%), H-11 → L+1 (4%), H-5 → L+7 (4%), H → L+3 (3%), H → L+9 (3%)
17	2.4025	H-35 → L (26%), H-18 → L (14%), H-35 → L+1 (3%), H-35 → L+6 (2%), H-19 → L (5%), H-15 → L (3%), H-14 → L (5%), H-12 → L (4%), H-11 → L (5%), H-10 → L (3%), H-1 → L+4 (4%)
18	2.4135	H-8 → L+7 (13%), H-8 → L+9 (44%), H-38 → L+9 (2%), H-9 → L+9 (3%), H-8 → L+3 (8%), H-8 → L+8 (8%), H-6 → L+11 (5%)
19	2.4142	H-9 → L+6 (12%), H-9 → L+8 (43%), H-9 → L+2 (8%), H-9 → L+9 (8%), H-8 → L+8 (3%), H-7 → L+10 (5%)
20	2.4161	H → L+2 (96%), H-2 → L+2 (3%)

Figure R10. Calculated energy levels and possible ISC channels of RuDA in A) monomeric, and B) dimeric forms. The arrows refer to the possible ISC channels.

- The size of the generated nanoparticles was characterized using DLS. While the plot shows a uniform distribution, more data points especially towards bigger particle sizes are necessary for a better understanding of the particle distribution. In addition, the polydispersity of the mixture needs to be determined.

Answer: Thank you for your professional comment and suggestion. Dynamic light scattering (DLS) measurement was carried out again. The particle sizes and polydispersity index (PDI) are presented in **Figure R11**. The related statements have been added to the revised manuscript.

Figure R11. DLS analysis and TEM image (inset) of RuDA-NPs.

- The photostability of RuDA was studied over a time frame of 0-25 min. In the experimental section it is noted that the whole absorption spectrum is monitored. Unfortunately, the spectrum is not shown and only a normalized value at one wavelength. The whole spectrum should be recorded and shown (maybe in the supporting information) to ensure that none of the other parts are changing. In addition, beside monitoring in dependence of the irradiation time also the light dose needs to be studied as a crucial factor. For a meaningful insight, the authors would need to investigate which light dose the here studied time frame corresponds to. They should ensure to reach at least the clinically applied levels (within the range of 200 J cm^{-2}).

Answer: Thank you for your suggestion. The whole spectrum from 190 to 1100 nm regarding the photostability of RuDA has been supplemented in the revision (**Figure S7**).

Besides, the appropriate laser power density, irradiation time and light dose are very important for efficient phototherapy. According to previous literature reports, different power densities and irradiation time of 808 nm laser were used for the study of phototherapy, including 2.0 W cm^{-2} for 30 min with 5 min break for each 10 min exposure (*Angew. Chem.* **2020**, 132, 21638-21643); 1.0 W cm^{-2} for 20 min (*Angew. Chem.* **2019**, 131, 7890-7894); 1.0 W cm^{-2} for 10 min (*Nat. Comm.* **2019**, 10, 1192; *Angew. Chem. Int. Ed.* **2019**, 58, 1638-1642; *J. Am. Chem. Soc.* **2017**, 139, 16235-16247; *Angew. Chem.* **2017**, 129, 13872-13876); 0.7 W cm^{-2} for 10 min (*Nat. Comm.* **2018**, 9, 2798); 0.5 W cm^{-2} for 10 min (*Angew. Chem. Int. Ed.* **2021**, 60, 6047-6054). Since the high power density is harmful to normal tissues as reported that "The desired power density was $200 - 500 \text{ mW cm}^{-2}$, regardless of spot size, light dose - $200 - 300 \text{ J cm}^{-2}$ during PDT session" (*Proc. of SPIE*, **2005**, 5973: 65-70), 0.5 W cm^{-2} with 10 min irradiation (300 J cm^{-2}) was chosen for the *in vivo* study. However, total light dose is over the range of clinically applied levels (200 J cm^{-2}). Thank you so much for your reminder. In the following study, we will take fully consideration of the laser power density, irradiation time and light dose.

- In general, the authors should mention in every biological experiment, the applied light dose as a crucial parameter

Answer: The light dose was supplemented in the revised experimental section.

- Ruthenium-Arene complexes are well-known to degrade within physiological conditions (see works of Dyson, Hartinger or Keppler). The stability of the reported compounds needs to be

investigated in-depth to ensure understanding which components cause which biological effects.

Answer: The stability of metal-based compounds is of great importance for their biological activity. Our group has tried to elucidate the underlying mechanism of metal-based anticancer agents based on their kinetic reactivity and electronic structures, e.g. platinum(II) compounds (*Chem. Eur. J.* **2014**, 20, 15216-15225; *J. Inorg. Biochem.* **2017**, 175, 20-28), platinum(IV) compounds (*Inorg. Chem.* **2017**, 56, 9851-9859), and Ru(II)-arene compounds (*Inorg. Chem.* **2018**, 57, 8396-8403). It is undoubted that Ru(II)-arene compounds are liable to hydrolyze under physiological conditions. However, through supramolecular assembly, most of the monomers are trapped inside the supramolecular bulk that is isolated from the environment or solvent with only a few molecules on the interface between the environment and the supramolecular bulk. Therefore, the hydrolysis rate of RuDA is very slow. Moreover, nanocarrier-loaded RuDA (RuDA-NPs) with Pluronic F127 as isolation carriers can keep RuDA from further hydrolysis, which can significantly improve their stability, thereby maintaining their photo-physicochemical properties (**Figure R2**).

- The authors refer to an inhibition effect upon irradiation of the photosensitizer. However, no specific cell mechanism is inhibited and instead cytotoxic species generated which cause cell death. This terminology needs to be corrected.

Answer: Thanks for your constructive suggestion. It has been demonstrated that mitochondrion is a main target of Ru(II)-arene complexes. To assess the mitochondrial dysfunction, we carried out JC-1 and MitoSOX Red staining were carried out to evaluate the mitochondrial membrane potential and superoxide generation capacities, respectively. As shown in **Figure R12**, intense green (JC-1) and red (MitoSOX Red) fluorescence is observed in both RuDA and RuDA-NPs-treated cells under 808 nm laser irradiation, demonstrating that RuDA and RuDA-NPs can induce mitochondrial membrane depolarization and superoxide generation effectively.

Figure R12. JC-1 and MitoSOX Red staining of MDA-MB-231 cells treated with RuDA-NPs (50 μ M) or RuDA (50 μ M) upon 808 nm laser (0.5 W cm^{-2}) irradiation for 10 min. Scale bars: 30 μ m.

The cell death mechanism was determined using flow cytometry based on annexin V-FITC/propidium iodide (PI) assay. As depicted in **Figure R13**, upon 808 nm irradiation, both RuDA and RuDA-NPs induced a significant increased incidence of the early stage apoptosis (lower right quadrant) in MDA-MB-231 cells compared with PBS- or PBS plus laser-treated cells. However, when Vc was added, the apoptotic rates of RuDA and RuDA-NPs were significantly reduced to 15.8% and 17.8% from 50.9% and 52.0%, respectively, confirming the essential role of ROS in the photocytotoxicity of RuDA-NPs and RuDA. Besides, negligible necrotic cells (upper left quadrant) were observed for all the test groups, demonstrating that apoptosis may be a major form of RuDA- and RuDA-NPs-induced cell death.

Figure R13. Flow cytometry analysis for apoptosis of MDA-MB-231 cells treated with RuDA-NPs (50 μ M) or RuDA (50 μ M) in the presence and absence of Vc (0.5 mM), and irradiated with or without 808 nm laser (0.5 W cm⁻²) for 10 min.

As oxidative stress injury is a major determinant of cell apoptosis, the expression of the nuclear factor erythroid 2-related factor 2 (Nrf2), a key regulator of the antioxidant system, was investigated in RuDA-NPs-treated MDA-MB-231 cells to elucidate the modes of action induced by RuDA-NPs under irradiation. Meanwhile, the expression of its downstream protein, heme oxygenase-1 (HO-1), was also assayed. As shown in **Figure R14**, RuDA-NPs-mediated phototherapy upregulated the expression levels of Nrf2 and HO-1 as compared with the PBS group, demonstrating that RuDA-NPs can stimulate the oxidative stress signaling pathway. Besides, the expression of heat-responsive heat shock protein Hsp70 was evaluated to explore the photothermal effect of RuDA-NPs. Obviously, the cells treated with RuDA-NPs plus 808 nm irradiation exhibited increased Hsp70 expression compared with the other two groups, reflecting the cellular responses to hyperthermia.

Figure R14. (A) Cellular Nrf-2, Hsp70, and HO-1 expressions of MDA-MB-231 cells treated with RuDA-NPs (50 μ M) with or without 808 nm laser irradiation (0.5 W cm^{-2} , 10 min, 300 J cm^{-2}). (B) Quantification of Nrf2, Hsp70, and HO-1 expressions of MDA-MB-231 cells; the intensities of Nrf2, Hsp70, and HO-1 were normalized to that of β -actin using Gel-Pro 32 software.

- The biodistribution of the metal complex is missing as a reference to the injected dose. The presented data in Figure 5A should be normalized towards the injected dose.

Answer: The biodistribution of Ru (% injected dose (ID) of Ru per gram of tissues) contents have been converted as a reference to the injected dose.

Figure R15. *Ex vivo* tissue distribution of RuDA-NPs injected through the tail vein at a concentration of 10.0 $\mu\text{mol kg}^{-1}$ determined in mice by the content of Ru (% injected dose (ID) of Ru per gram of tissues) at different post injection time.

- As an important factor, the subcellular localization as well cell death mechanism of the photosensitizer has a drastic effect on the photodynamic efficacy. For a full biological evaluation of the here reported nanoparticles, the sub-organelle localization and cell death mechanism needs to be investigated.

Answer: As you pointed out, the subcellular localization of the photosensitizers has a drastic effect

on the photodynamic efficacy, which correlates to their mechanisms of action. We have demonstrated that Ru(II)-based complexes are preferentially located in the nucleus and mitochondria. Therefore, the subcellular localizations of RuDA and RuDA-NPs were investigated. As shown in **Figure R16**, RuDA and RuDA-NPs exhibited a similar cellular distribution profile, with the highest accumulation in mitochondria (62.5 ± 4.3 and 60.4 ± 3.6 ng/mg protein, respectively). However, only a small amount of ruthenium was observed in the nuclear fraction for RuDA and RuDA-NPs (3.5% and 2.1%, respectively). The residual cellular fraction contained the remainder of the ruthenium, 31.7% (30.6 ± 3.4 ng/mg protein) for RuDA and 42.9% (47.2 ± 4.5 ng/mg protein) for RuDA-NPs, respectively. Overall, RuDA and RuDA-NPs were mainly accumulated in the mitochondria.

As oxidative stress injury is a major determinant of cell apoptosis, the expression of the nuclear factor erythroid 2-related factor 2 (Nrf2), a key regulator of the antioxidant system, was investigated in RuDA-NPs-treated MDA-MB-231 cells to elucidate the modes of action induced by RuDA-NPs under irradiation. Meanwhile, the expression of its downstream protein, heme oxygenase-1 (HO-1), was evaluated. As shown in **Figure R14**, RuDA-NPs-mediated phototherapy upregulated the expression levels of Nrf2 and HO-1 as compared with those in the other two groups, demonstrating that RuDA-NPs-mediated phototherapy can effectively stimulate the oxidative stress signaling pathway. Besides, photothermal effect of RuDA-NPs was evaluated by exploring the expression of heat-responsive heat shock protein Hsp70. Obviously, cells treated with RuDA-NPs plus 808 nm irradiation exhibited an increased Hsp70 expression compared with those in the other two groups, reflecting the cellular responses to hyperthermia.

Figure R16. Ruthenium uptake (ng/mg protein) in the different cellular compartments of MDA-MB-231 cells treated with RuDA or RuDA-NPs at 50 μ M for 12 h quantified by ICP-MS.

The reviewers' comments and suggestion are much constructive and helpful for us to improve the quality of our manuscript. We hope our explanation and revision can address the concern. We appreciate the reviewers so much for their precious time and professional effort on reviewing our manuscript.

REVIEWERS' COMMENTS

Reviewer #1 (Remarks to the Author):

The authors have properly addressed the questions and it is ready to be published.

Reviewer #2 (Remarks to the Author):

The manuscript entitled "A Highly Efficient Supramolecular Photosensitizer Derived from an Arene-Ru(II) Complex Self-assembly for Near-Infrared-Activated Photodynamic/Photothermal Therapy" submitted to Nature Communication by Zhao, Xia and Guo thematizes the development of Ru(II)-Arene complexes for near-infrared photodynamic and photothermal therapy. The authors have addressed the vast majority of scientific concerns raised by the reviewers. The only concern which has not been adequately addressed remains the stability of the metal complex and the particles (see below for more information). Despite these additional information and clarifications, this reviewer believes that this study lacks in novelty to become suitable for Nature Communication. The authors have recently published an article in Advances Functional Materials (Adv. Funct. Mater. 2021, 2008325) which describes analogous compounds. These compounds have been designed, chemically and biologically evaluated in the same manner. Even the obtained results are comparable. As such this concept and the here described results are expected and do not present novel research findings. The same aspect has been raised by Reviewer 1 during his/her evaluation. Therefore, I recommend rejection of the presented manuscript due to lack of novelty.

Within the revised version of the manuscript, the authors have investigated the stability of the metal complex by UV/VIS absorption spectroscopy. These experiments present a preliminary insight but are not sufficient to determine the stability of the metal complex in particular as the extended aromatic ligand is responsible for the NIR absorption. Meaningful understanding of the stability could exemplary be obtained by HPLC analysis following incubation in a biological environment. If the hypothesis is that the aggregation and encapsulation is enhancing the stability of the metal complex, the authors would need to provide compelling evidence for this. Exemplary this could be done by TEM images followed incubation in a biological environment.

Reviewer #1 (Remarks to the Author):

The authors have properly addressed the questions and it is ready to be published.

Answer: Thank you for your positive response and approval.

Reviewer #2 (Remarks to the Author):

The manuscript entitled "A Highly Efficient Supramolecular Photosensitizer Derived from an Arene-Ru(II) Complex Self-assembly for Near-Infrared-Activated Photodynamic/Photothermal Therapy" submitted to Nature Communication by Zhao, Xia and Guo thematizes the development of Ru(II)-Arene complexes for near-infrared photodynamic and photothermal therapy. The authors have addressed the vast majority of scientific concerns raised by the reviewers. The only concern which has not been adequately addressed remains the stability of the metal complex and the particles (see below for more information). Despite these additional information and clarifications, this reviewer believes that this study lacks in novelty to become suitable for Nature Communication. The authors have recently published an article in Advances Functional Materials (Adv. Funct. Mater. 2021, 2008325) which describes analogous compounds. These compounds have been designed, chemically and biologically evaluated in the same manner. Even the obtained results are comparable. As such this concept and the here described results are expected and do not present novel research findings. The same aspect has been raised by Reviewer 1 during his/her evaluation. Therefore, I recommend rejection of the presented manuscript due to lack of novelty.

Answer: Thank you so much for your comments. For your queries, we'd like to make the following explanations, and hope our explanations can address your concern.

Firstly, as you previously stated "the author have prepared a new Ru(II) complex with a significant absorption in the NIR region. As this is a highly desirable property, the authors should elaborate on their design of this metal complex so other scientists could learn from their design strategy to obtain the desired photophysical properties.", this series of metal-based photosensitizers exhibit unique photophysical and photochemical properties, including broad absorption in the NIR region, aggregation-induced $^1\text{O}_2$

generation, and synergistic PDT and PTT effects, which are different from the conventional metal-based photosensitizers and highly desired for phototherapy.

Secondly, although a NIR-triggered iridium(III) complex has been designed before (Adv. Funct. Mater. 2021, 2008325), the underlying mechanisms of the properties of such complexes remain unclear. In order to elucidate the underlying mechanisms of RuDA, the theoretical calculation was carried out to describe the excited state properties and behaviors. Besides, a novel control compound (RuET) was deliberately designed for comparison, and the photoelectrochemical properties of RuDA and RuET were then examined and compared using electrochemical impedance spectroscopy (EIS) and transient photocurrent measurements, which provides valuable information for elucidating the underlying mechanisms of RuDA. Our present study can provide useful insight into the molecular design of NIR-excited organometallic Ru(II)-arene photosensitizers.

Thirdly, organometallic Ru(II)-arene complexes have been widely explored as chemotherapeutic agents for cancer treatment owing to their low toxicity and facile modification. However, the application of Ru(II)-arene complexes in PDT is relatively rare. By taking advantages of the ionic character and pseudo-octahedral half-sandwich structure of organometallic Ru(II)-arene complex, we extends the application of Ru(II)-arene complexes not only in chemotherapy but also in NIR-activated phototherapy.

Within the revised version of the manuscript, the authors have investigated the stability of the metal complex by UV/VIS absorption spectroscopy. These experiments present a preliminary insight but are not sufficient to determine the stability of the metal complex in particular as the extended aromatic ligand is responsible for the NIR absorption. Meaningful understanding of the stability could exemplarily be obtained by HPLC analysis following incubation in a biological environment. If the hypothesis is that the aggregation and encapsulation is enhancing the stability of the metal complex, the authors would need to provide compelling evidence for this. Exemplary this could be done by TEM images followed incubation in a biological environment.

Answer: Thank you for your comments. The UV/Vis spectroscopy technique is an

effective method for the investigation of the kinetic properties of metal complexes, which can determine not only the stability but also the hydrolysis rates of metal complexes. For example, the technique has been used to investigate the hydrolysis rates and kinetic reactivity of metal complexes (e.g. organometallic Ru(II)-arene complexes, Pt(II) and Pt(IV) complexes) by our group (Inorg. Chem. 2018, 57, 8396-8403; Inorg. Chem. 2017, 56, 9851-9859; J. Med. Chem. 2015, 58, 6368-6377; Chem. Eur. J. 2014, 20, 15216 -15225). Besides, it has been well demonstrated that supramolecular assembly can improve stability of the chemically unstable species, such as indocyanine green (ICG) and perylene diimide (PDI) radical anion (Angew. Chem. Int. Ed. 2020, 59, 3793-3801).

According to your suggestion, high-performance liquid chromatography (HPLC) technique was used to determine the stability of RuDA and RuDA-NPs. As exhibited in **Figure R1**, RuDA is stable in the first hour in the mixture of methanol and water (50/50, v/v), and the hydrolysis was observed after 4 h. However, only a broad bump peak was observed for RuDA-NPs, demonstrating that HPLC technique is not suitable for the analysis of RuDA-NPs nanoparticles. Hence, gel permeation chromatography (GPC) was used to evaluate the stability of RuDA-NPs in the medium of PBS (pH = 7.4). As shown in **Figure R2**, negligible changes of the peak height, peak width and peak area of RuDA-NPs were observed after 8 h incubation under the test conditions, indicating the excellent stability of RuDA-NPs. In addition, TEM images demonstrated that the morphology of RuDA-NPs nanoparticles almost remained unchanged after 24 h in diluted PBS buffer (pH = 7.4, **Figure R3**).

Figure R1. HPLC chromatograms. RuDA (50 μ M) in a solution of methanol and water (50/50, v/v) after (A) 0 h, (B) 1 h, and (C) 4 h incubation.

Figure R2. GPC profiles. (A) RuDA-NPs (50 μ M) in PBS buffer (pH = 7.4) after different incubation times (0h, 1h, 8h). (B) Water.

Figure R3. TEM images. RuDA-NPs after incubation in PBS (pH = 7.4) for (A) 0 h, and (B) 24 h. A representative image of three independent tests from each group is shown.